# NanoSpec: Accelerating Speculative Decoding using Minimalist In-Context Vocabularies

Zhiyang Chen [1]   Daliang Xu [2]   Yinyuan Zhang [3]   Chenghua Wang [2]   Mengwei Xu [2]   Yun Ma [1]

## Abstract

The massive vocabulary sizes of large language models, often exceeding 100k tokens, impose a computational bottleneck on the final linear projection layer during speculative decoding. Existing vocabulary pruning solutions rely on static or coarsely-grained sub-vocabularies that necessitate large active sizes ($\sim$30k) to maintain draft quality. We propose `NanoSpec`, a novel training-free approach that breaks this trade-off by dynamically constructing a minimalist, context-aware active vocabulary for each generation step. Leveraging the inherent temporal locality of language generation, `NanoSpec` achieves high coverage while slashing the average vocabulary size by over $40\times$ (to $<$3k tokens) without requiring any auxiliary trained parameters. To realize the theoretical benefits of such high sparsity on modern hardware, we introduce a system-algorithm co-design that overcomes the inefficiencies of sparse memory access through asynchronous gathering and GPU-resident state management. As a complementary plug-and-play module, `NanoSpec` cuts draft time by an average of 51.6%, delivering a 1.17-1.29$\times$ end-to-end speedup over the state-of-the-art speculative decoding methods EAGLE-2 and EAGLE-3 across 7 tasks and outperforming complex training-based pruning baselines.

[1]Institute for Artificial Intelligence, Peking University, Beijing, China [2]State Key Laboratory of Networking and Switching Technology; School of Computer Science, Beijing University of Posts and Telecommunications, Beijing, China [3]Key Laboratory of High Confidence Software Technologies (Peking University), Ministry of Education; School of Computer Science, Peking University, Beijing, China. Correspondence to: Yun Ma <mayun@pku.edu.cn>, Daliang Xu <xudaliang@bupt.edu.cn>.

*Proceedings of the 43$^{rd}$ International Conference on Machine Learning*, Seoul, South Korea. PMLR 306, 2026. Copyright 2026 by the author(s).

## 1. Introduction

Large Language Models (LLMs) have demonstrated remarkable capabilities across diverse tasks (Han et al., 2021; Zhou et al., 2025; Naveed et al., 2025). To capture multilingual and domain-specific semantics (Tao et al., 2024; Takase et al., 2025), their vocabulary sizes frequently exceed 100k tokens (e.g., 128k for Llama-3, 152k for Qwen-2.5). While beneficial for generation quality, these expansive vocabularies pose significant challenges for efficient inference.

Speculative decoding (SD) (Miao et al., 2024; Chen et al., 2023; Leviathan et al., 2023) has emerged as a premier technique to accelerate inference without compromising quality, employing a fast draft mechanism to generate tentative sequences for parallel verification by the target LLM. The efficacy of SD relies fundamentally on the draft mechanism being significantly faster than the target. However, recent work exposes a critical bottleneck when scaling SD to large-vocabulary models: the computational cost of the draft model's final linear projection (the LM head) becomes prohibitive (Zhao et al., 2025; Weng et al., 2025; Zhang et al., 2025). Crucially, this bottleneck affects diverse draft architectures, including separate small models and integrated heads like EAGLE or Medusa (Li et al., 2024; Cai et al., 2024) as they all rely on the LM head to map the hidden states to massive vocabulary spaces. For example, this single projection step can consume over 60% of the total draft inference time on Llama-3.1-8B and Qwen-2.5-7B (Weng et al., 2025), severely limiting achievable speedups.

A direct remedy is to prune the draft model's output vocabulary. As draft tokens are verified by the target model whose vocabulary remains untouched, pruning does not compromise final output quality. Existing approaches typically adopt context-agnostic or coarse-grained strategies. Methods like FR-Spec (Zhao et al., 2025) and VocabTrim (Goel et al., 2025) statically select high-frequency tokens as the pruned vocabulary based on corpus statistics, while others like DynaSpec (Zhang et al., 2025) and CORAL (Weng et al., 2025) employ trained auxiliary routers to select from predefined sub-vocabulary clusters (Weng et al., 2025; Zhang et al., 2025). However, a fundamental limitation of these approaches is their inability to adapt fine-grainedly to immediate contexts. To maintain reasonable draft quality

*Table 1.* Comparison of vocabulary pruning strategies (Zhao et al., 2025; Zhang et al., 2025) for speculative decoding using Llama-3.1-8B-Instruct on SpecBench and HumanEval. Speed denotes generation speed (tokens/s); Acc. Len. denotes average acceptance length. `NanoSpec` achieves competitive acceptance across all baselines via dynamic vocabulary concentration (Section 4.2).

| Method | Pruning Strategy | Extra Training | Active Vocab. | EAGLE-2 | | EAGLE-3 | |
|---|---|---|---|---|---|---|---|
| | | | | Speed | Acc. Len. | Speed | Acc. Len. |
| Full Vocabulary | None | **None** | 128k | 336.9 | **3.80** | 362.2 | **4.25** |
| FR-Spec | High-Frequency Tokens | **None** | 32k | 369.7 | 3.62 | 396.8 | 3.96 |
| DynaSpec | Route to Fixed Clusters | 1 MLP | 27k | 367.7 | 3.73 | 388.3 | 4.08 |
| **NanoSpec (Ours)** | **Dynamic by Context** | **None** | $<3$k | **392.7** | 3.59 | **430.0** | **4.25** |

and capture necessary long-tail tokens, they are forced to operate with relatively large active vocabularies (e.g., ∼30k tokens as shown in Table 1), missing the opportunity for substantial acceleration.

In this work, we challenge this paradigm by seeking the minimal sufficient vocabulary required for each specific generation step. Our core insight is that language generation exhibits strong temporal locality (Saxena, 2023; Wang et al., 2025; Chen et al., 2025): the next token is overwhelmingly likely to be present in the immediate context or be a close extension of it. Driven by this, we propose `NanoSpec`[1], a training-free approach that dynamically constructs a minimalist active vocabulary (e.g., 2k-3k tokens) per step solely from token history and recent high-probability candidates, without relying on any learned routers. While theoretically compelling, realizing the benefits of a highly dynamic, ultra-small vocabulary presents its own system-level challenge: the resulting sparse memory access patterns for gathering LM head weights are notoriously inefficient on modern GPUs, potentially negating computational savings. To overcome this, `NanoSpec` integrates system-algorithm co-design through asynchronous gathering and GPU-resident state management, effectively mitigating the latency overhead of dynamic sparse computation.

As summarized in Table 1, `NanoSpec` achieves superior state-of-the-art generation speeds (430.0 tokens/s on Llama-3.1-8B with EAGLE-3 (Li et al., 2026)) using an average dynamic vocabulary of $<3$k tokens, an order of magnitude smaller than existing approaches (27k-32k), without requiring any additional training or auxiliary parameters. By leveraging the inherent contextual nature of language supported by optimized system design, we break the long-standing trade-off between vocabulary size and draft acceptance rate, unlocking a new regime of efficient speculative decoding.

In summary, our work makes the following contributions:

- We provide empirical analysis quantifying the untapped potential of dynamic vocabulary pruning and the strong

[1]Open-source code: https://github.com/csAugust/NanoSpec.

temporal locality in LLM generation.

- We propose a simple, training-free dynamic pruning method achieving high coverage with minimal vocabulary size ($<3$k), co-designed with system-level techniques to resolve sparse memory access overheads.

- We demonstrate `NanoSpec` serves as a plug-and-play module, achieving 1.17–1.29× complementary speedup over EAGLE-2 and 1.19–1.28× over EAGLE-3, surpassing complex trained baselines across diverse benchmarks.

## 2. Motivation

### 2.1. Rethinking Draft Vocabulary Pruning: Beyond the Accuracy-Efficiency Trade-off

Validated by recent studies (Zhao et al., 2025; Weng et al., 2025; Zhang et al., 2025), the output projection (LM head) of the draft model has emerged as a dominant bottleneck in speculative decoding, especially as model vocabularies inflate to over 100k text tokens (e.g., 128k of Llama-3, 152k of Qwen-2.5) (Tao et al., 2024). To mitigate this, pruning the draft vocabulary is a compelling strategy.

However, existing state-of-the-art approaches operate under a *static paradigm*: either selecting a fixed subset of high-frequency tokens (e.g., FR-Spec (Zhao et al., 2025), Vocab-Trim (Goel et al., 2025)) or training a router to select from a fixed set of pre-clustered vocabularies (e.g., CORAL (Weng et al., 2025), DynaSpec (Zhang et al., 2025)). These methods seem trapped in an inevitable dilemma: a smaller vocabulary size is necessary for compute savings, but blindly restricting it severely degrades the draft's acceptance rate due to failure of recalling long-tail tokens, often neutralizing any speed gains. We argue that this perceived trade-off stems from the inherent limitations of their static or coarsely-grained nature, which forces them to ignore fine-grained, instance-specific context. To quantify this trade-off and the resulting untapped potential, we define two scenarios:

- *Oracle* Scenario (Theoretical Limit): we hypothesize an ideal state where the draft model's acceptance length remains unaffected by vocabulary reduction. Formally,

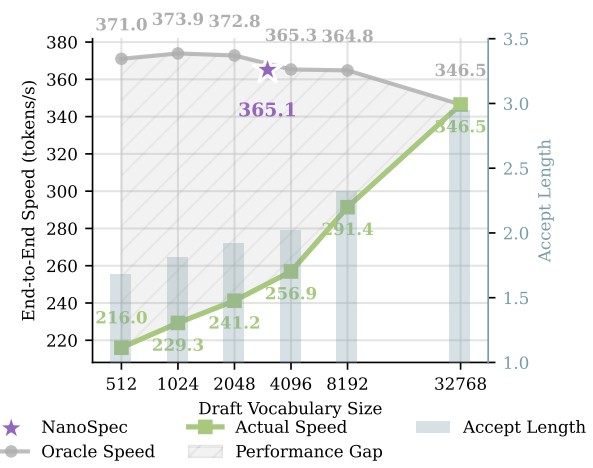

(a) End-to-End Speed vs. Vocab Size

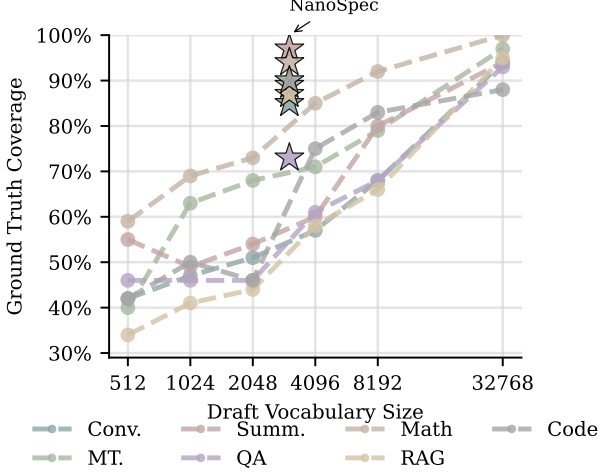

(b) Ground Truth Coverage Empirical Analysis

*Figure 1.* Motivation analysis using Llama-3.1-8B. (a) The untapped potential in static vocabulary pruning. While theoretical acceleration exists for smaller vocabularies (oracle speed), actual speed drops rapidly due to reduced draft acceptance rates. Our target is to bridge this gap. (b) Vocabulary coverage analysis across diverse domains. Static method (dashed lines) requires large vocabularies for high coverage. Our approach (`NanoSpec`, stars) achieves superior coverage with minimalist dynamic vocabulary (maximum of 3k tokens).

let $\bar{\alpha}^V$ denote the average acceptance length under the full vocabulary $V$, $T_{verify}$ the target model's verification latency, and $T_{draft}^{V'}$ the draft latency with reduced vocabulary $V'$. The oracle speed is defined as:

$$\text{Speed}_{oracle}^{V'} = \frac{\bar{\alpha}^V}{T_{verify} + T_{draft}^{V'}}. \quad (1)$$

This isolates and highlights the pure computational gain achievable by shrinking the LM head's computation.

- *Actual* Scenario: we evaluate the real-world performance using FR-Spec (Zhao et al., 2025) as a representative static pruning method, where acceptance length drops as the pruned vocabulary shrinks.

We vary the pruned vocabulary sizes from 32k down to 0.5k to explore the potential speedup, where 32k is the optimal setting empirically reported in FR-Spec. Figure 1(a) illustrates this analysis on Llama-3.1-8B-Instruct with a case on multi-turn conversation tasks.

The oracle line (gray) shows that theoretically, shrinking the vocabulary from 32k down to roughly 2k should yield continuous speed gains, though extremely small dimensions below 1k exhibit minor non-monotonicity due to GPU underutilization at such small GEMM sizes. However, the actual line (green) reveals a stark reality: static pruning yields rapidly diminishing returns below a critical threshold (~16k) and suffers catastrophic performance drops at smaller sizes (<2k). This sharp decline occurs because overly aggressive, context-agnostic pruning fails to capture long-tail tokens crucial for the specific current context, dras-

tically reducing the draft acceptance rate, as shown in the blue bars of Figure 1(a). The substantial shaded area between the actual and theoretical curves represents a massive performance gap, the untapped potential of speculative decoding currently masked by inefficient, static vocabulary management. This gap is the direct target of our work.

### 2.2. The Bridge: Leveraging Strong Temporal Locality

Our approach is rooted in a key insight designed to bridge this gap: the behavior of LLM generation is governed by strong *temporal locality* (Saxena, 2023; Wang et al., 2025). We hypothesize that the tokens LLMs generate next are overwhelmingly likely to be present in the recent context history or are closely related extensions of it. Based on this hypothesis, we argue that the active vocabulary of the draft model can be dynamically pruned per step to a much smaller, context-aware subset while capturing sufficient correct semantics.

To validate this empirically, we analyze the ground truth coverage: the probability that the correct token $y_t^*$ selected by the target LLM at each step $t$ is present in a pruned vocabulary. We compare two strategies:

- *FR-Spec* (Zhao et al., 2025): Static vocabulary with top-$K$ high-frequency tokens based on corpus statistics.

- `NanoSpec`: Dynamic vocabulary based on the current context and generation trajectory (detailed in Section 3).

Figure 1(b) compares these coverage rates across varied domains (detailed in Section 4.1) using Llama-3.1-8B-Instruct.

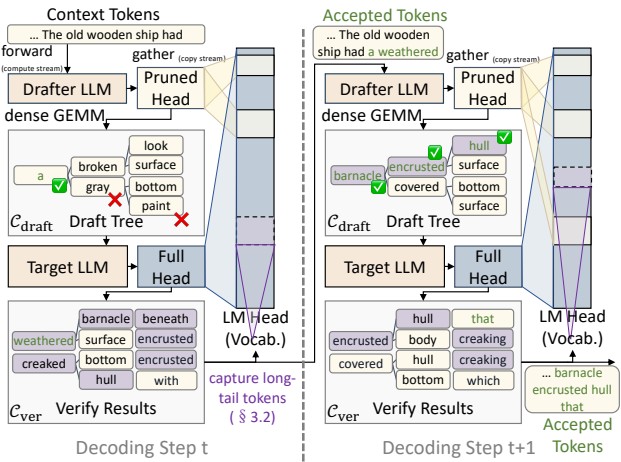

*Figure 2.* Overview of `NanoSpec`. Given the context "The old wooden ship had", long-tail tokens "weathered, barnacle, hull" lie outside a fixed boundary, despite being highly probable in this specific context. Consequently, the draft model is forced to select generic, suboptimal high-frequency alternatives ("broken, surface"). `NanoSpec` recovers from this by dynamically constructing the vocabulary of draft model.

The static methods (dashed lines) require prohibitively large vocabularies (often $> 16k$ or even $> 32k$) to achieve acceptable coverage ($> 85\%$). This perfectly explains the performance sharp drop observed in Figure 1(a): below a certain size, static vocabularies simply miss too many correct tokens. In contrast, `NanoSpec` (stars) consistently achieves high coverage (73%-97% depending on the task) while maintaining an exceedingly small average vocabulary size of a maximum of 3k tokens.

This analysis provides strong motivating evidence: a minimalist, dynamically constructed vocabulary based on explicit context is sufficient to capture the vast majority of generated tokens. This pivotal insight allows us to operate in the high-speed ($< 3k$ size) regime that was previously unattainable by static methods, effectively enabling us to recover the untapped potential identified in Figure 1(a).

## 3. Methodology

### 3.1. Preliminaries and Problem Formulation

Consider a standard auto-regressive large language model $M_\theta$. At decoding step $t$, given context $x_{<t}$, the model predicts the next token distribution via a final linear projection layer, conventionally termed the LM head. Let $W_{head} \in \mathbb{R}^{V \times d}$ represent the weight of the LM head, where $V$ is the vocabulary size originally defined during pretraining, and $d$ is the hidden dimension. The logits $z_t \in \mathbb{R}^V$ are computed from the final hidden state $h_t \in \mathbb{R}^d$ as:

$$z_t = W_{head} h_t. \tag{2}$$

Speculative decoding (Leviathan et al., 2023; Miao et al., 2024; Chen et al., 2023; Li et al., 2024) employs a smaller, faster draft model $M_{draft}$ to generate a draft of $\gamma$ speculative tokens, which are subsequently verified by the target LLM $M_{target}$ in parallel. While effective, the theoretical speedup is heavily bounded by the latency of the draft model itself. As vocabulary sizes $V$ in modern LLMs scale massively (e.g., $V \approx 128k$ for Llama-3, $V \approx 152k$ for Qwen-2.5), the projection in Eq. (2) becomes surprisingly expensive, reaching a complexity of $O(V \cdot d)$ per token (or $O(V \cdot d \cdot b)$ when computing $b$ draft tokens in parallel via GEMM). For efficient draft models where the backbone consists of few layers (e.g., Medusa heads or EAGLE layers (Cai et al., 2024; Li et al., 2024)), this final linear projection often dominates total inference latency, severely capping the achievable speedup of existing approaches (Zhao et al., 2025; Weng et al., 2025).

Our objective is to accelerate this operation by replacing the draft model's full vocabulary set indices $\mathcal{I}_{full} = \{1, \ldots, V\}$ with a substantially smaller, dynamically determined subset of indices $\mathcal{I}_t \subset \mathcal{I}_{full}$ at each decoding step $t$, such that $|\mathcal{I}_t| \ll V$. This allows replacing the dense computation in Eq. (2) with a sparse operation involving only a sub-matrix $W_{head}[\mathcal{I}_t, :]$. Crucially, as we only optimize drafting and do not alter the target model's vocabulary, the generation quality is guaranteed to be lossless (i.e., identical to auto-regressive decoding (Leviathan et al., 2023)).

### 3.2. Algorithm: Context-Aware Dynamic Vocabulary Construction

As outlined in Section 2, we propose a deterministic mechanism to construct a context-aware $\mathcal{I}_t$ by leveraging the strong temporal locality inherent in LLM generation. Our core insight is that the optimal draft vocabulary at any moment is heavily skewed towards tokens present in the immediate context history and high-probability candidates recently considered by the target model.

Figure 2 illustrates how `NanoSpec` updates its dynamic vocabulary. We formalize this by defining an evolving *candidate token stream* and deriving $\mathcal{I}_t$ via a fixed-size sliding window over this stream.

**Candidate Stream Initialization (Prefill Phase).** Given an input prompt sequence $x_{1:L}$, the target model processes the prompt to generate a sequence of logits $Z_{1:L} = (z_1, \ldots, z_L)$. We define a Top-K operator $\mathcal{T}_K(z)$ that returns the set of indices corresponding to the $K$ largest values in a logit vector $z$. The candidate stream $\mathbf{S}$ is initialized as the concatenation of the exact prompt tokens and the union of top-$K_{pre}$ candidates from each position in the prompt:

$$\mathbf{S}^{(0)} = (x_1, \ldots, x_L) \oplus \text{tuple}\left(\bigcup_{i=1}^{L} \mathcal{T}_{K_{pre}}(z_i)\right), \tag{3}$$

where hyper-parameter $K_{pre}$ controls exploration breadth during prefill and $\oplus$ denotes sequence concatenation.

**Dynamic Stream Update (Decoding Phase).** At any decoding step $t$, let the draft model generate a speculative token tree, and let the target model's verification process on the validated branch yield final logits $z_{verify}^{(t)}$. We collect two sets of new candidates to append to the stream:

1. **Draft Candidates ($\mathcal{C}_{draft}^{(t)}$):** The set of all unique token indices present anywhere in the generated draft tree at step $t$, regardless of acceptance.

2. **Verify Candidates ($\mathcal{C}_{ver}^{(t)}$):** The top-$K_{ver}$ high-probability tokens according to the target model's distribution at this step, i.e., $\mathcal{C}_{ver}^{(t)} = \mathcal{T}_{K_{ver}}(z_{verify}^{(t)})$. $K_{ver}$ is a hyper-parameter controlling per-step exploration.

The candidate stream is updated by appending the sequence of these new indices:

$$\mathbf{S}^{(t+1)} = \mathbf{S}^{(t)} \oplus \text{tuple}(\mathcal{C}_{draft}^{(t)}) \oplus \text{tuple}(\mathcal{C}_{ver}^{(t)}). \quad (4)$$

**Vocabulary Construction via Sliding Window.** To ensure bounded memory usage and computational efficiency while retaining the most relevant history, we impose a strict size limit $W_{max}$ on the active vocabulary. The dynamic vocabulary index set $\mathcal{I}_t$ for the next step is formally defined as the unique set of the most recent $W_{max}$ tokens in the stream:

$$\mathcal{I}_t = \text{Unique}(\text{Suffix}(\mathbf{S}^{(t)}, W_{max})), \quad (5)$$

where $\text{Suffix}(\cdot, W_{max})$ returns the last $W_{max}$ elements of the sequence, and $\text{Unique}(\cdot)$ extracts unique elements. This construction naturally implements a recency-based eviction policy, where tokens whose most recent occurrence falls outside the window are discarded.

## 3.3. System: Hardware-Accelerated Dynamic Vocabulary Realization

While the dynamic vocabulary theoretically reduces FLOPs by a factor of roughly $V/|\mathcal{I}_t|$, realizing this speedup on modern GPUs is non-trivial. Implementing NanoSpec naively requires matrix computation on non-contiguous rows from $W_{head}$ corresponding to the dynamic indices $\mathcal{I}_t$. This random, sparse memory access pattern breaks GPU memory coalescing, resulting in severe bandwidth underutilization that often negates theoretical gains. To address this, we propose a hardware-aware implementation tailored to the specific requirements of our context-aware algorithm.

**Avoiding Sparse Computation via Asynchronous Gathering.** Instead of performing inefficient sparse computations, our strategy is to efficiently transform the sparse data access into a dense computation: we maintain a pre-allocated buffer in GPU memory, dynamically gather the weights of new active tokens from $W_{head}$ and pack them contiguously into the active buffer at each step. This approach enables computing $W_{head}[\mathcal{I}_t, :]$ using a dense matrix.

To eliminate blocking caused by $M_{draft}$ waiting for packed weights, we further pipeline copy stream with execution of model backbone by compute stream. Specifically, as the indices $\mathcal{I}_{t+1}$ required for the next draft step are only known after the target model completes verification at step $t$ (generating $\mathcal{C}_{ver}^{(t)}$), we overlap the memory-intensive weight gathering for step $t+1$ with the compute-intensive draft execution for step $t+1$. We utilize dual CUDA streams and fine-grained event synchronization:

- **Compute Stream:** executes the main neural network layers (backbone) of both $M_{draft}$ and $M_{target}$.

- **Copy Stream:** executes specialized kernels that gather required weights from global memory into a pre-allocated, contiguous dense buffer.

As soon as $\mathcal{I}_{t+1}$ is determined via the GPU-resident state update (described below), we launch the draft model's backbone computation on the compute stream and simultaneously launch the weight gather kernel on the copy stream. A CUDA event is recorded after the gather operation. The compute stream is made to wait on this event only immediately before executing the draft's LM head projection.

This pipeline design ensures that the gathering latency is effectively hidden behind the draft backbone computation. By the time the draft backbone finishes, the requisite weights ($W_{head}[\mathcal{I}_{t+1}, :]$) are continuously packed in the buffer, allowing performing LM head projection with a highly efficient dense matrix multiplication.

**Eliminating Overhead via GPU-Resident State Management.** The sliding window mechanism necessitates high-frequency updates to the active token set $\mathcal{I}_t$ at every generation step. Managing this dynamic state on the CPU involves prohibitive host-device synchronization overhead, which would introduce pipeline bubbles.

To eliminate this overhead, NanoSpec implements fully GPU-resident state management. The active sliding window states are maintained entirely in VRAM using bitmaps. We employ highly parallel, lock-free CUDA kernels to identify unique new candidates from draft trees and verification results, pushing them into the stream and updating the active indices set directly on the device. This near-zero-overhead management ensures that the dynamic logic of our algorithm does not become a new system bottleneck.

# 4. Evaluation

## 4.1. Experimental Setup

**Benchmarks.** We employ the SpecBench benchmark (Xia et al., 2024), which is specifically designed to evaluate speculative decoding methods across a diverse set of six tasks: machine translation (MT.) (Bojar et al., 2014), multi-turn conversation (Conv.) (Zheng et al., 2023), RAG and QA (RAG, QA) (Kwiatkowski et al., 2019), mathematical reasoning (Math) (Cobbe et al., 2021), summarization (Summ.) (Nallapati et al., 2016). We further augment it with HumanEval (Chen, 2021) for code generation (Code) tasks. The maximum generation length is set to 1024 tokens and the greedy sampling is adopted for all tasks.

**Baselines.** We compare NanoSpec against auto-regressive decoding and two categories of baselines:

- *SD backbones*: EAGLE-2 (Li et al., 2024) and EAGLE-3 (Li et al., 2026), where computing the full LM head serves as the full-vocabulary baseline. Since existing pruning methods are exclusively evaluated on EAGLE-2, we adopt it as the primary backbone and additionally report EAGLE-3 results.

- *Vocabulary pruning methods*:
  - FR-Spec (Zhao et al., 2025): Static method that selects a fixed high-frequency token subset based on corpus statistics.
  - DynaSpec (Zhang et al., 2025): Router-based method that clusters vocabularies offline and trains an MLP router to dynamically select active clusters.

**Implementation.** Our system is built on top of the open-source FR-Spec framework, with custom modifications to the Python and CUDA kernels to realize the proposed techniques and support EAGLE-3. Since DynaSpec is not open-sourced and several critical hyper-parameters are undisclosed, we re-implemented its router by training an 8.6M-parameter MLP with 256 vocabulary clusters (via spherical K-Means) on ShareGPT following the methodology described in their paper.

**Models, Hyper-Parameters and Hardware.** We evaluate on Llama-3.1-8B-Instruct, Llama-3.2-1B-Instruct and Qwen-2-7B-Instruct. All baselines use official EAGLE-2 checkpoints as the draft model with identical hyper-parameters: draft tree depth of 5 and maximum draft tokens of 60. For EAGLE-3, we train our own draft models with full-vocabulary LM heads for NanoSpec, as the official EAGLE-3 checkpoints use a statically pruned sub-vocabulary ($\sim$32k tokens), which is incompatible with dynamic vocabulary methods. FR-Spec uses pruned vocabulary size 32k (its reported optimum). For NanoSpec, we set $K_{pre} = K_{ver} = 3$ and $W_{max} = 3072$. All experiments are conducted on a single NVIDIA H20 GPU.

## 4.2. End-to-End Speedup

We first evaluate the end-to-end generation throughput (tokens/s, including both prefill and decoding) on both EAGLE-2 (Table 2) and EAGLE-3 (Table 3) backbones. Results of Qwen-2 are detailed in Appendix A.2.

**Results on EAGLE-2** (Table 2, Figure 3). On Llama-3.1-8B-Instruct, NanoSpec achieves an average speed of **392.7 tokens/s** (**2.25**× over AR), surpassing router-based DynaSpec (2.13×) and static FR-Spec (2.12×). The advantage is more pronounced on Llama-3.2-1B-Instruct, where the drafting overhead dominates. Standard EAGLE-2 with full-vocabulary computation offers negligible gains over AR (1.05×), while NanoSpec achieves **1.35**×, substantially outperforming FR-Spec (1.17×) and DynaSpec (1.15×).

**Results on EAGLE-3** (Table 3). NanoSpec achieves **1.19**× speedup over vanilla EAGLE-3 on Llama-3.1-8B (430.0 vs. 362.2 tokens/s) and **1.28**× on Llama-3.2-1B (944.8 vs. 739.6 tokens/s), consistently outperforming FR-Spec. Notably, on context-heavy tasks (Summarization, RAG), NanoSpec achieves comparable or higher acceptance lengths than full-vocabulary EAGLE-3. We attribute this to the noise-filtering effect of our dynamic vocabulary: removing irrelevant tokens from the draft vocabulary causes softmax renormalization to concentrate probability mass onto context-relevant candidates, producing a draft distribution that better aligns with the target model and thus improves acceptance probability.

## 4.3. Average Acceptance Length

We analyze the average acceptance length, which quantifies the mean number of draft tokens verified as correct by the target model per step, as a hardware-independent assessment of drafting accuracy.

On EAGLE-2 (Table 2), NanoSpec achieves acceptance lengths of 3.59 (8B) and 3.11 (1B), highly competitive with the full-vocabulary baseline (3.80 and 3.16) and significantly outperforming static FR-Spec on the 1B model (2.70), where static pruning suffers severe degradation. On EAGLE-3 (Table 3), NanoSpec maintains an average acceptance length of 4.25 (8B) and 3.39 (1B), closely matching the full-vocabulary EAGLE-3 baseline (4.25 and 3.45) while FR-Spec drops more significantly to 3.96 and 3.27. These results confirm that our training-free dynamic mechanism effectively identifies vital tokens that static methods miss, maintaining high drafting quality without auxiliary parameters. The slight acceptance length trade-off against full-vocabulary baselines is more than compensated by the substantial reduction in draft computation time.

*Table 2.* Comparison on end-to-end generation speed (tokens/s) and acceptance length across tasks and models. The baseline is standard auto-regressive decoding (AR). Avg. Speed is the mean across tasks (averaged over 3 runs), with relative speedup over AR in parentheses. Avg. Len. denotes the mean acceptance length per step. The best results for each model are bolded.

| Model | Method | MT | Conv | RAG | Math | QA | Summ | Code | Avg. Speed | Avg. Len. |
|---|---|---|---|---|---|---|---|---|---|---|
| Llama-3.1 8B-Instruct | AR | 175.7 | 178.5 | 165.0 | 178.2 | 174.2 | 174.3 | 162.3 | 172.6 (1.00×) | 1.00 |
| | EAGLE-2 | 311.2 | 380.2 | 300.2 | 403.2 | 316.4 | 315.0 | 332.0 | 336.9 (1.93×) | 3.80 |
| | FR-Spec | 338.4 | 419.3 | 330.2 | 456.5 | 351.0 | 356.0 | 336.3 | 369.7 (2.12×) | 3.62 |
| | DynaSpec | 334.8 | 417.9 | 323.9 | 440.8 | 340.6 | 343.6 | 372.4 | 367.7 (2.13×) | 3.73 |
| | **NanoSpec** | **349.5** | **440.2** | **352.1** | **480.0** | **361.3** | **385.1** | **381.1** | **392.7 (2.25×)** | 3.59 |
| Llama-3.2 1B-Instruct | AR | 675.6 | 677.2 | 624.7 | 677.9 | 672.2 | 661.7 | 671.3 | 665.8 (1.00×) | 1.00 |
| | EAGLE-2 | 657.0 | 777.3 | 653.8 | 812.3 | 644.4 | 608.3 | 753.4 | 700.9 (1.05×) | **3.16** |
| | FR-Spec | 751.2 | 844.7 | 715.2 | 913.5 | 718.5 | 672.8 | 835.9 | 778.8 (1.17×) | 2.70 |
| | DynaSpec | 729.1 | 826.3 | 688.4 | 865.7 | 684.3 | 637.3 | 908.5 | 762.8 (1.15×) | 2.92 |
| | **NanoSpec** | **850.6** | **1008.2** | **830.4** | **1069.3** | **822.0** | **796.2** | 928.5 | **900.7 (1.35×)** | 3.11 |

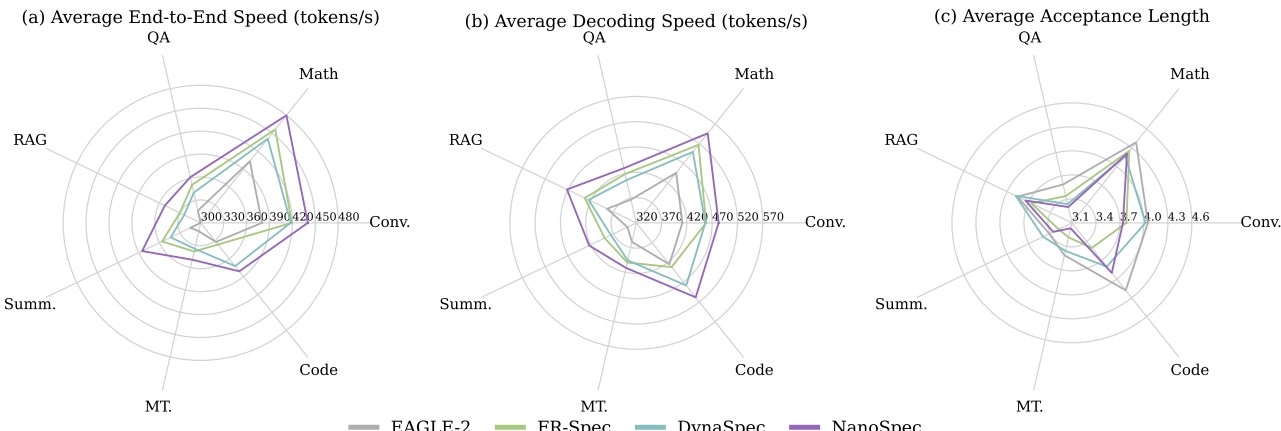

*Figure 3.* Performance comparison across tasks on Llama-3.1-8B-Instruct. Despite having shorter acceptance lengths, `NanoSpec` consistently outperforms all baselines in terms of both end-to-end speed and decoding speed.

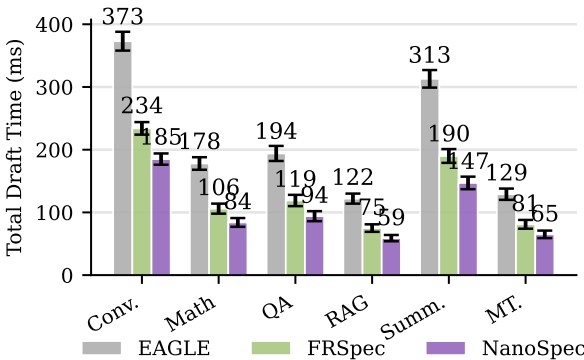

*Figure 4.* Draft time comparison. `NanoSpec` effectively reduces draft overhead with a smaller vocabulary. The error bars indicate the minimum and maximum of average draft time.

### 4.4. Draft Efficiency Analysis

We investigate the computational overhead during the drafting phase to understand where the end-to-end speedup originates. Figure 4 compares the total drafting time across tasks on Llama-3.1-8B-Instruct. Full-vocabulary EAGLE-2 suffers from significant latency as it computes against the entire 128k vocabulary at every drafting step. FR-Spec mitigates this via static pruning, but still requires a large active vocabulary (32k) to capture long-tail tokens. `NanoSpec` achieves the lowest overhead, reducing drafting time by 51.6% compared to EAGLE-2 and 20.3% compared to FR-Spec.

To further pinpoint the source of this reduction, we profile the per-step wall-clock latency using CUDA event recording (Table 4). The draft backbone computation remains constant across all methods (∼1.4 ms), confirming that the speedup originates entirely from the LM head. `NanoSpec` reduces LM head latency from 2.330 ms (full vocabulary) to 0.237

*Table 3.* Compatibility with EAGLE-3: end-to-end speed (tokens/s) and acceptance length across tasks. `NanoSpec` delivers consistent speedups over vanilla EAGLE-3 without additional training.

| Model | Method | MT | Conv | RAG | Math | QA | Summ | Code | Avg. Speed | Avg. Len. |
|---|---|---|---|---|---|---|---|---|---|---|
| Llama-3.1 8B-Instruct | EAGLE-3 | 308.6 | 412.4 | 332.5 | 440.9 | 322.0 | 339.6 | 379.7 | 362.2 (1.00×) | **4.25** |
| | FR-Spec | 345.5 | 456.8 | 365.0 | 501.7 | 359.5 | 388.4 | 360.6 | 396.8 (1.10×) | 3.96 |
| | **NanoSpec** | **356.7** | **489.2** | **393.0** | **530.2** | **381.3** | **442.3** | **417.5** | **430.0 (1.19×)** | **4.25** |
| Llama-3.2 1B-Instruct | EAGLE-3 | 654.4 | 805.1 | 675.5 | 881.9 | 671.8 | 616.7 | 871.6 | 739.6 (1.00×) | **3.45** |
| | FR-Spec | 817.8 | 987.5 | 817.3 | 1121.5 | 827.9 | 775.1 | 950.1 | 899.6 (1.22×) | 3.27 |
| | **NanoSpec** | **837.7** | **1029.0** | **846.7** | **1176.4** | **849.1** | **810.7** | **1064.0** | **944.8 (1.28×)** | 3.39 |

*Table 4.* Per-step latency breakdown (ms) of the draft model on Llama-3.1-8B-Instruct. The weight gather overhead of `NanoSpec` is fully hidden behind the concurrent backbone computation.

| Component | Full Vocab | FR-Spec | NanoSpec |
|---|---|---|---|
| Draft Backbone | 1.405 | 1.401 | 1.403 |
| Weight Gather | – | – | 0.003 |
| Draft LM Head | 2.330 | 0.771 | 0.237 |
| **Total Draft Step** | 3.735 | 2.172 | **1.640** |

*Table 5.* Ablation study on average end-to-end speed across all SpecBench tasks using Llama-3.1-8B-Instruct.

| Vocab. Source | Async. Gather | Avg. Speed |
|---|---|---|
| Ctx. Only | ✓ | 313.3 |
| Ext. Only | ✓ | 351.4 |
| Ctx. + Ext. | × (Indexed GEMM) | 364.1 |
| **Ctx. + Ext.** | ✓ | **394.6** |

ms, a **9.8×** reduction. Crucially, the weight gather overhead introduced by our dynamic vocabulary is entirely absorbed by the concurrent backbone computation on the other CUDA stream, validating our asynchronous dual-stream design. Overall, the total draft step latency drops by 56.1% compared to full-vocabulary EAGLE-2 and 24.5% compared to FR-Spec, directly translating to the substantial end-to-end speedups reported earlier.

### 4.5. Ablation Study

To examine the individual contributions of our proposed components, we conduct an ablation study on Llama-3.1-8B-Instruct by comparing four configurations:

- Ctx. Only: The active vocabulary $\mathcal{I}$ is restricted to tokens from the input prompt and prefill-phase candidates ($\mathbf{S}^{(0)}$). No new candidates are added during decoding.
- Ext. Only: The vocabulary consists solely of draft tree

*Table 6.* VRAM overhead of `NanoSpec` on Llama-3.1-8B (FP16).

| Buffer | Purpose | Size |
|---|---|---|
| `token_ids` | GPU-resident token set index (i.e., $\mathcal{I}_t$) | 16 KB |
| `tokens_tensor` | Temporarily store new tokens (i.e., $\mathcal{C}_{draft}^{(t)}, \mathcal{C}_{ver}^{(t)}$) | 12 KB |
| `repack_buf` | Dense buffer that repacks LM-head weight | 24.00 MB |
| **Total** | | **24.02 MB** |

tokens and verification top-$K$ candidates ($\mathcal{C}_{draft} \cup \mathcal{C}_{ver}$), ignoring the prompt context.

- Ctx. + Ext. (× Async.): Full dynamic vocabulary with a naive Indexed GEMM kernel for sparse computation, without asynchronous gathering.

- Ctx. + Ext. (✓ Async.): Our complete method with both full dynamic vocabulary and system-level optimization.

As shown in Table 5, Ctx. Only yields 313.3 tokens/s, while Ext. Only improves to 351.4 tokens/s, demonstrating the importance of dynamic updates. Combining both sources further improves to 364.1 tokens/s, confirming that capturing both prompt-derived global context and generation-derived local continuity is essential. The naive indexed GEMM approach (×Async.) is bottlenecked by non-contiguous memory access. Applying asynchronous gathering (✓Async.) boosts speed from 364.1 to 392.7 tokens/s, highlighting the necessity of hardware-aware system design.

### 4.6. Memory Overhead Analysis

The GPU-resident state management of `NanoSpec` introduces a deterministic memory overhead that is *independent of context length* ($O(1)$ w.r.t. sequence length). Table 6 details the additional VRAM consumption for Llama-3.1-8B ($d = 4096$, $W_{max} = 3072$, FP16).

The total of 24.02 MB is merely 0.03% of the model's 14GB+ memory pool. With 50% GPU memory allocated

to KV cache, the maximum sequence capacity decreases by only 0.04%. The overhead scales linearly with batch size but remains negligible compared to the KV cache that grows $O(N)$ with context length.

Additional analyses are provided in the appendix: robustness on larger vocabulary (Appendix A.2), hyper-parameter sensitivity (Appendix A.3) and analysis on vocabulary evolution (Appendix A.4).

## 5. Discussion

**Scaling Boundaries.** As the target model scales to 70B+, the verification latency $T_{\text{verify}}$ (dominated by KV-cache access and target compute) grows to dominate end-to-end latency, causing the relative proportion of draft overhead $T_{\text{draft}}$ to shrink. By Amdahl's Law, even fully eliminating $T_{\text{draft}}$ yields diminishing returns when verification is the bottleneck. The primary impact zone of `NanoSpec` therefore lies in edge devices and moderate-scale LLMs (1B–8B), where inflated vocabularies (128k+ tokens) make the draft LM head a severe bottleneck.

**Deployment in Serving Frameworks.** `NanoSpec` modifies only the draft model's LM head computation and maintains all state on GPU, making it orthogonal to framework-level optimizations such as PagedAttention (Kwon et al., 2023) and RadixAttention (Zheng et al., 2024). Integration into production frameworks (e.g., vLLM, SGLang) requires only injecting the asynchronous gather kernels into the draft worker process, with no modifications to the target model, KV cache or scheduler.

## 6. Related Work

**Speculative Decoding.** Speculative decoding (Miao et al., 2024; Chen et al., 2023; Leviathan et al., 2023) has emerged as a prominent technique for accelerating auto-regressive LLM inference. It employs a smaller, faster draft model to tentatively generate draft sequences, which are then verified in parallel by the larger target model. EAGLE-2 (Li et al., 2024) uses a single Transformer layer as the draft model, and EAGLE-3 (Li et al., 2026) further scales it against distributional shift. Gumiho (Li et al., 2025a) explores combining Transformer layer and MLP layer. RSD (Liao et al., 2025) invokes verification dynamically through a reward model. Recent work (Timor et al., 2025) also proposes augmenting the vocabulary of target model for better performance. Our optimization is complementary with these works as they all rely on LM head to compute logits and suffer from inefficiencies with the LLM vocabulary scaling.

**Vocabulary Pruning for Speculative Decoding.** To mitigate the LM head bottleneck in speculative decoding, recent work focuses on pruning the draft model's vocabulary. We categorize existing approaches into two groups: (1) *Frequency-based methods*: FR-Spec (Zhao et al., 2025) and VocabTrim (Goel et al., 2025) restrict the draft LM head to a fixed subset of high-frequency tokens derived from corpus statistics. While simple and training-free, they are context-insensitive and struggle with long-tail tokens, leading to lower acceptance rates in diverse scenarios. (2) *Router-based methods*: DynaSpec (Zhang et al., 2025) and CORAL (Weng et al., 2025) train auxiliary router networks to select from predefined token clusters at each step. However, these learned approaches still rely on statically preclustered vocabularies, resulting in non-negligible training overhead. In contrast, `NanoSpec` is the first fully dynamic, context-aware and training-free approach that achieves state-of-the-art speedups with an order-of-magnitude smaller vocabulary (<3k).

**General Vocabulary Pruning.** Beyond speculative decoding, vocabulary pruning is employed in other LLM domains for different objectives, such as early-exiting (Vincenti et al., 2024; Xu et al., 2025) and reinforcement learning (Li et al., 2025b; Hashimoto & Tsuruoka, 2019) to improve efficiency. Other work proposes pruning for inference to reduce memory consumption (Keith et al., 2024; Nozaki et al., 2025). A critical distinction is that these methods apply pruning that directly affects the final output or training objective, often incurring a trade-off in accuracy. As our work applies pruning solely to the draft model, the speculative decoding framework ensures the final output is rigorously verified by the target model, guaranteeing lossless outputs identical to standard auto-regressive decoding.

## 7. Conclusion

In this paper, we challenged the prevailing paradigm of using static or complex trained routers for vocabulary pruning in speculative decoding, proposing `NanoSpec`, a training-free dynamic method with system-level co-design to break the long-standing trade-off between vocabulary size and acceptance rate. The empirical evaluation demonstrates that `NanoSpec` achieves state-of-the-art end-to-end speedups with an unprecedentedly small vocabulary of <3k tokens, delivering consistent acceleration across diverse tasks, models and speculative decoding algorithms.

## Acknowledgements

This work was supported by National Natural Science Foundation of China under the grant number 62595734, Beijing Natural Science Foundation under the grant number L253005, the Beijing Municipal Science & Technology Commission and the Administrative Commission of Zhongguancun Science Park.

## Impact Statement

This work aims to improve the inference efficiency of large language models, particularly focusing on reducing the bottleneck caused by large vocabularies. The primary potential benefits of our approach lie in improving computational resource utilization and promoting accessibility. By achieving significant speedups without requiring additional training or complex auxiliary models, our method lowers the barrier toward deploying capable LLMs in resource-constrained environments. Furthermore, increased inference efficiency can contribute to reduced energy consumption per generated token, supporting more environmentally sustainable AI practices. While faster text generation inherently applies to both beneficial and potentially harmful use cases of LLMs, the focus of this contribution is strictly on infrastructural efficiency optimization.

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

---

**Algorithm 1** `NanoSpec`: Speculative Decoding with Dynamic Minimalist In-Context Vocabulary

---

1: **Input:** Target Model $M_{target}$, Draft Model $M_{draft}$, Prompt $x_{1:L}$, Max Steps $T$, Max Draft Tokens $\gamma$.
2: **Output:** Prediction $x_{L:T}$
3: **Hyper-parameters:** Prefill Top-K $K_{pre}$, Verify Top-K $K_{ver}$, Window Size $W_{max}$.
4: $h_{target} \leftarrow M_{target}.\text{forward}(x_{1:L})$
5: $Z_{1:L} \leftarrow M_{target}.\text{lm\_head}(h_{target})$
6: $\mathbf{S} \leftarrow (x_{1:L}) \oplus \bigcup_{i=1}^{L} \text{TopK}(Z_i, K_{pre})$ {Init Candidate Stream}
7: $\mathcal{I} \leftarrow \text{Unique}(\text{Suffix}(\mathbf{S}, W_{max}))$ {Init Dynamic Vocab Indices}
8: $W_{pruned} \leftarrow \text{Gather\_Async}(M_{draft}.W_{head}, \mathcal{I})$ {Stream 2: Copy pruned LM Head weights}
9: **for** $t = L$ to $T$ **do**
10:     **while** $j = 0 < \gamma$ **do**
11:         $h_{draft} \leftarrow M_{draft}.\text{forward}(x_{<t+j})$ {Stream 1: Compute}
12:         Sync stream 1 (compute) with stream 2 (copy)
13:         $z'_{draft} \leftarrow \text{GEMM}(h_{draft}, W^T_{pruned})$ {Stream 1: Dense GEMM on pruned LM Head}
14:         $x_{draft}, \text{num\_draft} \leftarrow \text{SelectDraftTokens}(z'_{draft}, \mathcal{I})$ {Drafts are sampled from pruned vocabulary}
15:         Append $x_{draft}$ to draft tree
16:         $j \leftarrow j + \text{num\_draft}$
17:     **end while**
18:     $h_{target} \leftarrow M_{target}.\text{forward}(x_{t:t+\gamma})$
19:     $z_{target} \leftarrow \text{GEMM}(h_{target}, W^T_{head})$ {Stream1: GEMM on full LM Head}
20:     $x_{new}, \text{num\_accept} \leftarrow \text{VerifyDraftTokens}(x_{t:t+\gamma}, z_{target})$ {Drafts are verified using full vocabulary}
21:     Append $x_{new}$ to context $x$
22:     $t \leftarrow t + \text{num\_accept} + 1$
23:     **if** predict end token **then**
24:         **break**
25:     **end if**
26:     $\mathcal{C}_{draft} \leftarrow \text{Unique}(\text{TokensInDraftTree}())$
27:     $\mathcal{C}_{ver} \leftarrow \text{TopK}(z_{target}, K_{ver})$
28:     $\mathbf{S} \leftarrow \mathbf{S} \oplus \mathcal{C}_{draft} \oplus \mathcal{C}_{ver}$ {Update stream $\mathbf{S}$ and $\mathcal{I}$ on device}
29:     $\mathcal{I}_{next} \leftarrow \text{Unique}(\text{Suffix}(\mathbf{S}, W_{max}))$
30:     $W_{pruned} \leftarrow \text{Gather\_Async}(M_{draft}.W_{head}, \mathcal{I}_{next})$ {Stream 2: Copy}
31: **end for**

---

# A. Appendix

### A.1. Pseudo-Code of **NanoSpec**

Algorithm 1 provides the complete pseudo-code for `NanoSpec`, illustrating the control logic of speculative decoding and the asynchronous streams for computation and memory gathering.

### A.2. Robustness Analysis on LLM with Larger Vocabulary

To investigate the robustness of `NanoSpec` when producing text from a larger vocabulary space, we analyze its performance on the Qwen-2-7B-Instruct model, which possesses a vocabulary size of approximately 152k (compared to 128k for Llama-3). The hyper-parameters of `NanoSpec` are the same with the settings in Section 4.1. The results of DynaSpec are from data reported in their paper.

Table 7 reports the average acceptance length across diverse tasks, which directly reflects the quality of the draft model's generation and the coverage capability of the pruned vocabulary. The results demonstrate that despite Qwen-2's massive vocabulary, `NanoSpec`, operating with a highly constrained maximum dynamic range of only $W_{max} = 3072$ tokens, maintains a high acceptance length (3.48), comparable to baselines that utilize much larger static vocabularies (i.e., 3.44 of FR-Spec) or full vocabulary spaces (i.e., 3.65 of EAGLE-2). This provides strong empirical evidence for the robustness of our context-driven, training-free approach in handling large-scale vocabulary models.

*Table 7.* Average acceptance length per step on Qwen-2-7B-Instruct across different tasks. A higher value indicates better quality of speculative generation and effective vocabulary coverage. The results of DynaSpec are from its reported optimal performance.

| Method | Average Acceptance Length by Task | | | | | | | Overall |
|---|---|---|---|---|---|---|---|---|
| | MT | Conv | RAG | Math | QA | Summ | Code | |
| **EAGLE-2 (Full Vocabulary)** | 3.03 | 3.86 | 3.70 | 4.31 | 3.06 | 3.43 | 4.19 | 3.65 |
| **FR-Spec** | 2.94 | 3.60 | 3.58 | 4.14 | 3.00 | 3.31 | 3.52 | 3.44 |
| **DynaSpec** | 2.86 | 3.72 | 3.32 | 4.18 | 2.97 | 3.24 | 3.96 | 3.46 |
| `NanoSpec` | 2.69 | 3.70 | 3.64 | 4.14 | 2.86 | 3.37 | 3.98 | 3.48 |

## A.3. Hyper-parameter Sensitivity Analysis

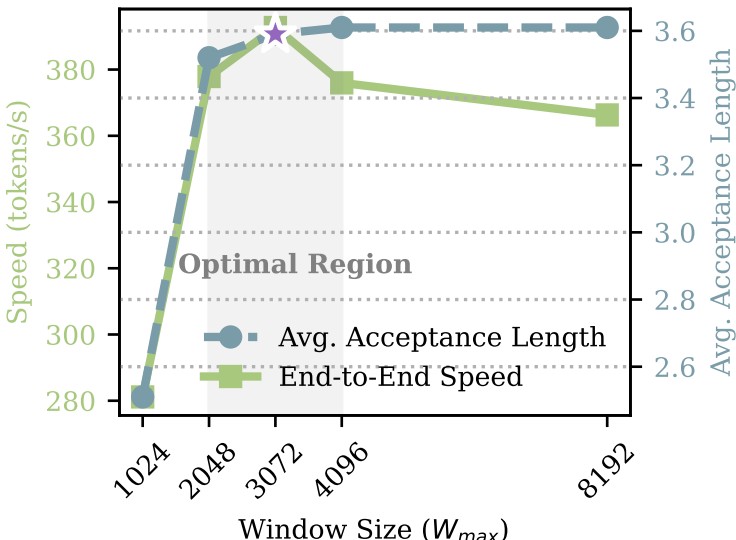

*Figure 5.* Sensitivity analysis of the maximum of vocabulary size ($W_{max}$). The dual-axis plot shows the trade-off between average acceptance length (dashed blue, right axis) and end-to-end Speed (solid green, left axis) as $W_{max}$ varies. The star marks our chosen default setting $W_{max} = 3072$.

The maximum size of the dynamic vocabulary ($W_{max}$) is a critical hyper-parameter in `NanoSpec`. It governs a fundamental trade-off between draft quality (vocabulary coverage, measured by acceptance length) and compute overhead (the latency associated with gathering weights and performing GEMM operations for a larger vocabulary). To define the operational boundaries of our approach, we conduct a sensitivity analysis by varying $W_{max}$ from 1024 to 8192 on SpecBench using Llama-3.1-8B-Instruct. The results are visualized in Figure 5.

**Analysis of Draft Quality.** The dashed blue line in Figure 5 illustrates the impact of $W_{max}$ on the average acceptance length. We observe a sharp increase in acceptance length as the window size grows from 1024 to 2048. However, beyond $W_{max} \approx 2048$, the acceptance length rapidly saturates and plateaus around a value of 3.6. This trend provides strong empirical evidence for our core insight regarding temporal locality: a relatively small, historically relevant context window is sufficient to capture the vast majority of tokens required for efficient speculative generation. Increasing the window size further yields diminishing returns in terms of vocabulary coverage.

**Analysis of System Performance.** The solid green line indicates the end-to-end generation speed. Initially, the speed increases alongside accepted length, peaking within the shaded optimal region ($2048 \leq W_{max} \leq 4096$). Crucially, as $W_{max}$ increases further beyond this region (e.g., up to 8192), the generation speed begins to decline, despite the acceptance length remaining stable. This decline highlights the diminished benefits of increasing vocabulary size under a length-specific context: as most generation tasks produce fewer than 4096 tokens in total, the larger sliding window is rarely fully utilized

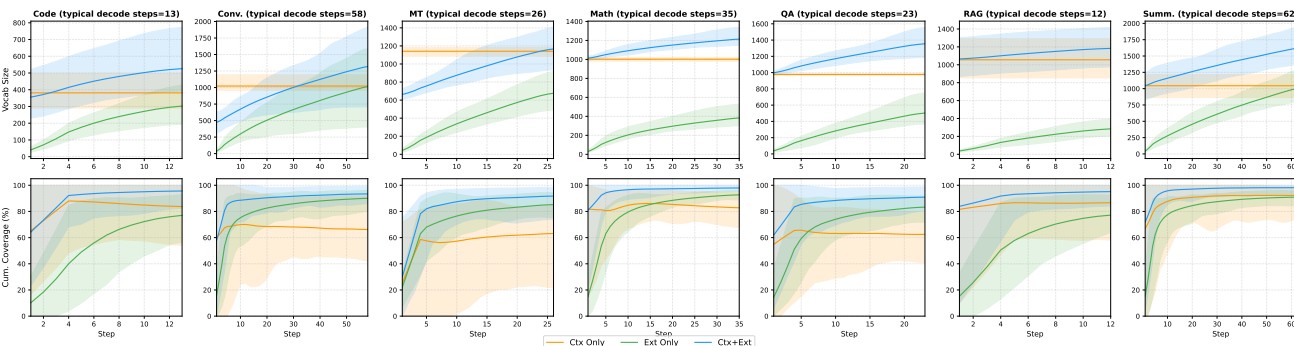

*Figure 6.* Per-category vocabulary evolution during decoding on Llama-3.1-8B-Instruct. Top row: active vocabulary size; bottom row: cumulative coverage (%). Solid lines denote the mean across samples; shaded regions indicate the min–max range. Each column corresponds to one task, with its typical decode length shown in the subtitle.

while the system overhead of computing the LM head continuously increases.

Consequently, an optimal operating region exists where the vocabulary is sufficiently large to maximize acceptance rate while staying small enough to minimize system overhead. We thus select $W_{max} = 3072$ (marked with stars in Figure 5) as our default robust setting, striking an effective balance between high algorithmic efficiency and minimal system latency.

### A.4. Dynamic Vocabulary Analysis

We analyze how different vocabulary sources contribute to coverage and how the active vocabulary evolves during decoding on Llama-3.1-8B-Instruct. Table 8 reports the aggregate coverage and acceptance length for each source configuration.

*Table 8.* Contribution of vocabulary sources to coverage and acceptance length on Llama-3.1-8B-Instruct (averaged across SpecBench tasks).

| Vocabulary Source | Coverage (%) | Avg. Accept Length |
|---|---|---|
| Ctx. Only | 64.23 | 2.89 |
| Ext. Only | 82.37 | 3.55 |
| **Ctx. + Ext. (Full `NanoSpec`)** | **91.65** | **3.59** |

To reveal the temporal dynamics behind these aggregate numbers, Figure 6 visualizes the per-category evolution of active vocabulary size (top row) and cumulative coverage (bottom row) over decoding steps for each source configuration, with shaded regions indicating the min–max range across samples.

**Complementarity of Ctx and Ext sources.** The two vocabulary sources exhibit complementary temporal profiles. Ctx Only (orange) provides a fixed vocabulary from the prompt, yielding high initial coverage on tasks that heavily reuse input text (e.g., RAG, Summarization reach ~80–90% from step 1) but degrading steadily as the generation diverges from the prompt. On open-ended tasks such as Conversation, Ctx Only coverage stagnates around 60–70%. Ext Only (green) starts near 0% coverage but ramps up rapidly within 5–10 steps as the candidate stream accumulates draft and verification tokens. The combined Ctx+Ext configuration (blue) inherits the strengths of both: it achieves high coverage from the very first step and sustains >90% coverage throughout decoding across all tasks.

**Vocabulary size saturation.** The top row of Figure 6 shows that under the full Ctx+Ext configuration, the active vocabulary grows monotonically but rapidly decelerates. For long-form tasks (Conv., Summ., >50 steps), the vocabulary plateaus between 1500 and 2000 unique tokens; for shorter tasks (Code, RAG), it remains below 1000. In all cases, the active vocabulary stays well below the $W_{max} = 3072$ capacity limit, explaining why the FIFO eviction overhead is negligible in practice.

**Code task behavior.** Code generation exhibits high overall coverage (Figure 1(b)) but moderate end-to-end speedups (Table 2). Figure 6 reveals the cause: code outputs are short (typical decode length of only 13 steps), and the Ext component requires ~5 steps to warm up. This initial low-coverage phase constitutes a large fraction of total generation, limiting the compounding benefits of high steady-state coverage that longer tasks enjoy.

