# OpenReview forum: "NanoSpec: Accelerating Speculative Decoding using Minimalist In-Context Vocabularies"
_ICML.cc/2026/Conference — ICML 2026 regular_

### Official Review · Reviewer_YHgy · 2026-03-09

**Soundness:** 2
**Presentation:** 2
**Significance:** 2
**Originality:** 3
**Overall Recommendation:** 4
**Confidence:** 4

**Summary:**

This paper presents NanoSpec, an approach to accelerate speculative decoding by significantly reducing the vocabulary size evaluated by the draft model's projection layer. Rather than relying on static pruning or training-based sub-vocabularies, the authors exploit the temporal locality of text generation—the tendency for subsequent tokens to be related to recently generated ones or those with high probabilities. To achieve this, the method continuously tracks a small, shifting set of candidate tokens (under 3,000) drawn from the input prompt, recent draft tokens, and top verification candidates. Since computing over a sparse, dynamic index set can lead to poor memory access patterns on hardware, the work is accompanied by system-level optimizations, including asynchronous fetching of model weights via dual CUDA streams and keeping the state updates entirely within GPU memory to avoid communication delays with the CPU. Evaluated on SpecBench + HumanEval across Llama-3-8B, Llama-3.2-1B, and Qwen-2-7B, NanoSpec achieves 1.12–1.32× end-to-end speedup over the state-of-the-art EAGLE-2 baseline and outperforms trained pruning methods (CORAL, DynaSpec) without any auxiliary parameters or training.

**Compliance With Llm Reviewing Policy:**

Affirmed.

**Final Justification:**

The authors have addressed my concerns, thus I am raising my score from 3 to 4.

**Key Questions For Authors:**

## Questions for Authors

1. **Regarding VRAM Consumption:** While the GPU-resident state management (using bitmaps) and pre-allocated buffers effectively hide latency, exactly how much additional VRAM do these structures consume? Could you provide a quantitative memory overhead analysis, particularly as batch sizes and context lengths scale?
2. **Regarding the "Oracle" Calculation in Figure 1:** How exactly is the "Oracle" speed (Tokens Per Second) calculated in Figure 1(a)? Furthermore, why evaluate a theoretical oracle using end-to-end TPS instead of directly plotting the upper bound of acceptance length or theoretical FLOP reduction? Could you provide the exact formulation used for this curve?
3. **Regarding Estimated Baselines:** Since the implementations for CORAL and DynaSpec are unavailable, estimating their TPS based on relative speedups from other papers is questionable. Could you either evaluate all methods in an identical setting or remove these estimated numbers to ensure a fair scientific comparison?
4. **Regarding Targeted Application Scenarios:** In what specific scenarios is this extreme pruning essential? For example, is NanoSpec primarily designed to benefit on-device deployments (where memory bandwidth is paramount) rather than general-purpose server-side deployment?

**Limitations:**

yes

**Strengths And Weaknesses:**

## Strengths

* **Elegantly Simple and Training-Free Design:** The core algorithmic idea — leveraging temporal locality to construct a context-aware dynamic vocabulary via a sliding window over a candidate token stream — is remarkably simple and requires zero additional training or auxiliary parameters. This makes it immediately deployable as a plug-and-play module on top of any existing speculative decoding system (e.g., EAGLE-2). The simplicity is a genuine strength; it is a clean, principled approach.
* **Strong Motivation and Methodology (Oracle vs. Actual):** The motivation section (Section 2, Figure 1(a)) employs an excellent methodology by comparing an "Oracle" scenario against an "Actual" scenario to visualize the untapped potential and performance gap in static vocabulary pruning. This clearly motivates why a new dynamic approach is needed.
* **Broad Applicability of Temporal Locality (Originality):** Utilizing the concept of temporal locality to build an extremely compact (<3k), dynamic vocabulary per-step is a creative and highly original perspective. This specific mechanism could easily inspire broader applications and caching schemes in general language generation beyond just speculative decoding.

## Weaknesses

* **Missing VRAM/Memory Overhead Analysis:** The paper proposes GPU-resident state management (maintaining sliding window states entirely in VRAM using bitmaps) and pre-allocated dense buffers to avoid CPU-GPU synchronization. However, there is no quantitative analysis of the extra VRAM footprint this requires. As sequence lengths and batch sizes scale, it is crucial to understand memory trade-offs, since extraneous VRAM usage directly competes with serving larger batches.
* **Unclear Definition of "Oracle" Speed in Motivation:** While Figure 1(a) effectively uses an "Oracle" scenario to motivate the problem, the paper entirely lacks a formal definition or formula for how this "Oracle speed (Tokens Per Second)" is calculated. It is unclear why a theoretical limit is plotted in end-to-end TPS rather than theoretical acceptance length or pure FLOPs reduction, making the exact quantification of this "untapped potential" difficult to verify.
* **Incomplete Baseline Comparison (Estimated Results):** A critical methodological concern is that CORAL and DynaSpec results in Table 2 are not reproduced. Since the implementations or trained models for CORAL and DynaSpec are unavailable, their speed data presented in figures and tables are estimated based on the relative speedups compared to the EAGLE baseline reported in their respective papers. Projecting end-to-end TPS across different papers is highly questionable, as differences in hardware, software versions, batch sizes, or measurement methodology can easily distort relative rankings. The authors should compare the methods in an identical setting or choose not to report these estimated numbers.

---

> ### Author Rebuttal · Authors · 2026-03-30
>
> We sincerely thank you for your thoughtful feedback. We are deeply encouraged by your recognition of NanoSpec's simple zero-training design, strong motivation and high originality of applying temporal locality. Below we address your concerns.
>
> ### 1. VRAM Consumption & Memory Overhead (W1, KQ1)
>
> NanoSpec's VRAM overhead is deterministic, extremely small and **independent of context length ($O(1)$)**. Maintaining the temporal sliding window only requires:
> 1. **Bitmap:** 16 KB (for a 128K vocabulary).
> 2. **Pre-allocated Dense Buffer:** $3000 \times H \times 2\text{ bytes (FP16)}$ ($H$ is hidden dim). For LLaMA-3-8B ($H=4096$), this is only $\sim$ 24 MB.
>
> To validate this, we empirically measured the exact VRAM footprint beyond vanilla EAGLE-2 for the LLaMA-3-8B model (FP16, hidden_dim=4096, maximum window size=3072). Our implementation of NanoSpec strictly adds only three pre-allocated GPU buffers:
>
> | Buffer Name             | Purpose                                | Overhead     |
> | :---------------------- | :------------------------------------- | :----------- |
> | `context_token_ids`     | GPU-resident token set index           | 16 KB        |
> | `context_tokens_tensor` | Temporarily store new tokens (i.e., $C_{draft}, C_{ver}$)         | 12 KB        |
> | `repack_buffer`         | Dense buffer for LM-head weight repack | 24.00 MB     |
> | **Total Overhead**      |                                        | **24.02 MB** |
>
> Our empirical tests match the theoretical analysis, showing an absolutely negligible 24.02 MB footprint. In our actual system test allocating 50% of the GPU memory to KV Cache, the maximum available sequence token budget marginally shifted from 422,724 tokens down to 422,538 tokens (**-0.04% reduction**). The total overhead represents at most **0.03%** of the entire 14GB+ memory pool required for a 7B/8B model context.
>
> ### 2. "Oracle" Speed Definition & Metric Choice (W2, KQ2)
>
> As shown by the bars in Figure 1(a) of the paper, shrinking the vocabulary normally degrades the actual acceptance length. Oracle Speed theoretically assumes the acceptance length never drops and remains identical to the full-vocabulary baseline ($\alpha_{full}$), whilst still fully benefiting from the pruned LM-head's latency reduction:
> $$Speed_{oracle} = \frac{\alpha_{full}}{T_{verify} + T_{draft\_{pruned}}}$$
>
> We plotted end-to-end (E2E) TPS instead of theoretical FLOP reduction because in memory-bound decoding, FLOP reduction does not translate linearly to latency reduction. E2E TPS is the only comprehensive metric that captures the actual runtime trade-off between draft speedup and acceptance loss.
>
> ### 3. Strictly Hardware-Aligned Baseline Evaluation (W3, KQ3)
>
> To avoid cross-paper estimation and strictly align hardware setups, we fully re-implemented DynaSpec's dynamic router. Following their paper, we clustered the vocabulary into 256 groups via spherical K-Means and trained an 8.6M-parameter MLP router on SharedGPT. Evaluated identically on our setup:
>
> | Model | Method | Speed (tokens/s) | Accept Length |
> | :--- | :--- | :--- | :--- |
> | LLaMA-3.2-1B| DynaSpec| 762.8 | 2.92 |
> | | **NanoSpec**| **900.7** | **3.11** |
> | LLaMA-3.1-8B | DynaSpec| 367.7 | **3.73** |
> | | **NanoSpec**| **392.7** | 3.59 |
>
> While undisclosed closed-source hyper-parameters (e.g., exact MLP configs, amount of clusters, learning rate) might slightly cap DynaSpec's reproduction optimum, these controlled empirical measurements firmly confirm that DynaSpec's heavy router computation severely bottlenecks wall-clock speedups. Conversely, NanoSpec's parameter-free locality entirely avoids this routing latency.
>
> ### 4. Application Scenarios & Scaling Constraints (KQ4)
>
> Your intuition is perfectly accurate. NanoSpec is optimal for edge/on-device deployments and moderate-scale LLMs, where the massive draft LM-head is the primary latency bottleneck.
>
> Let $\bar{\alpha}$ be the average accept length, and $T_{ar}$ the standard autoregressive time:
> $$S \approx \frac{\bar{\alpha} + 1}{T_{draft} / T_{ar} + T_{verify} / T_{ar}}$$
> For massive target models (e.g., 70B+), $T_{verify}$ (dominated by target KV-cache/compute fetching) increases immensely. Thus, even if NanoSpec flawlessly minimizes $T_{draft}$ to near-zero, the total speedup $S$ intrinsically diminishes as the massive target model overhead bounds Amdahl's Law.
>
> ---
>
> We are fully committed to incorporating all theoretical formalizations, exact empirical VRAM measurements, strictly reproduced baselines and scaling boundaries into the final manuscript. We hope these comprehensive explanation resolve your concerns and sincerely ask if you might consider raising the score. If you have any further questions, please let us know. We would be more than happy to provide further clarifications.

---

> > ### Author Rebuttal · Reviewer_YHgy · 2026-04-01
> >
> > My concern for Q1 and Q3 are fully resolved and I appreciate the efforts authors put into this rebuttal.
> >
> > For Q2, I understand conceptually why you are using oracle speed, but I think it is improper to call it e2e tps. In my understanding, e2e strongly implies a metric that is physically measured in real-world experiments, rather than derived from a theoretical formula. I suggest renaming this to "Theoretical Upper Bound TPS" in the final manuscript to avoid confusing readers.
> >
> > For Q4: Although this will not negatively affect my final rating, I would like to discuss the factors governing speedup in
> > speculative decoding as model size scales. The rebuttal suggests that the ratio $T_{verify}$ increases as target model sizes grow (e.g., 70B+). However, while absolute $T_{verify}$ naturally increases for larger models, $T_{ar}$ also scales up. Is it strictly true empirically or in theory that the ratio factor ($T_{verify} / T_{ar}$) also increases as model size grow?

---

> > > ### Author Response · Authors · 2026-04-01
> > >
> > > Thank you for your constructive follow-up and for acknowledging the resolution of Q1 and Q3. We sincerely appreciate the opportunity to continue this discussion. Below, we address your remaining questions regarding Q2 and Q4.
> > >
> > > ### Q2 (Naming of "e2e TPS")
> > >
> > > We fully agree with your suggestion. The term "e2e TPS" can indeed be misleading, as it may be interpreted as a physically measured wall-clock throughput. In the final manuscript, we will adopt a more precise descriptor such as "Theoretical Upper-Bound TPS" or "Oracle-Assisted TPS". We will explicitly provide its calculation formula and clarify that this metric represents a derived theoretical limit under ideal conditions (assuming zero system overhead), distinguishing it from empirically measured end-to-end throughput.
> > >
> > > ### Q4 (Scaling of $T_{\text{verify}}/T_{\text{ar}}$ with Model Size)
> > >
> > > We acknowledge that providing new large-scale empirical results (e.g., for 70B+ models) before the rebuttal deadline is not feasible due to time constraints. However, we are pleased to offer a detailed theoretical analysis to explore this question with you.
> > >
> > > First, we consider a simplified compute-bound scenario, ignoring memory-access overhead. For a decoder-only Transformer with $L$ layers, hidden size $h$, context length $n$, and a batch of $k$ tokens to verify in parallel:
> > > - An autoregressive step scales as $T_{\text{ar}} \sim L(c_1 h^2 + c_2 n h)$, where $c_1 h^2$ corresponds to the MLP and attention projections (parameters $\propto h^2$), and $c_2 n h$ accounts for attention over the KV-cache.
> > > - A verification step scales as $T_{\text{verify}} \sim L(c_1 k h^2 + c_2 k n h + c_3 k^2 h)$. The first two terms scale linearly with $k$ (parallel forward passes), while $c_3 k^2 h$ arises from attention among the $k$ speculated tokens.
> > >
> > > Thus, the ratio scales as:
> > > $$
> > > \frac{T_{\text{verify}}}{T_{\text{ar}}} = \frac{k(c_1 h^2 + c_2 n h) + c_3 k^2 h}{c_1 h^2 + c_2 n h} \xrightarrow[h \to \infty]{} k
> > > $$
> > >
> > > Under this pure compute-bound assumption, the ratio $T_{\text{verify}}/T_{\text{ar}}$ converges to the constant $k$ as $h$ grows large. **However, in practice, the actual latency trend is heavily governed by hardware characteristics**, particularly memory bandwidth versus compute utilization. As model size increases, the verification step could become increasingly memory-bound due to the massive KV-cache transfers required. In such regimes, $T_{\text{verify}}$ can grow disproportionately faster than $T_{\text{ar}}$. An empirical indication of this trend can be found in Sequoia [1] (Figure 8), which reports a larger $\frac{T_{\text{verify}}}{T_{\text{ar}}}$ ratio for Llama2-13B than for Llama2-7B.
> > >
> > > **Regarding the implication for our work:** Our core argument is not that the ratio $\frac{T_{\text{verify}}}{T_{\text{ar}}}$ monotonically increases with model size in all settings. Rather, we highlight a bottleneck shift: as the target model scales (e.g., to 70B+), the absolute magnitude of $T_{\text{verify}}$ frequently becomes the dominant component in the speculative decoding latency. Consequently, even if the draft stage is perfectly optimized ($T_{\text{draft}} \to 0$), the overall speedup $S$ becomes fundamentally constrained by the $T_{\text{verify}}/T_{\text{ar}}$ term, which is a direct reflection of Amdahl’s Law. This context underscores the value of NanoSpec, which is designed to minimize $T_{\text{draft}}$ to near-zero, ensuring that the draft phase does not become an additional bottleneck, especially in settings where draft cost is significant (e.g., on-device or moderate-scale deployments).
> > >
> > > We hope this theoretical elaboration clarifies our position. Thank you again for this engaging discussion, which has helped us refine the presentation of our work. We would be very grateful if you could consider raising your final rating based on the clarifications and planned revisions outlined above.
> > >
> > > [1] Chen, Zhuoming, et al. "Sequoia: Scalable and robust speculative decoding." NIPS 2024.

---

### Official Review · Reviewer_zr7X · 2026-03-10

**Soundness:** 3
**Presentation:** 3
**Significance:** 3
**Originality:** 3
**Overall Recommendation:** 4
**Confidence:** 4

**Summary:**

This paper studies draft-side vocabulary pruning for speculative decoding. The main idea is to replace the full draft vocabulary with a very small dynamic active vocabulary constructed from prompt tokens, recent draft candidates, and target-side verification candidates. The paper also proposes a systems implementation based on asynchronous gathering and GPU-resident state management to make this dynamic pruning efficient in practice.

**Compliance With Llm Reviewing Policy:**

Affirmed.

**Final Justification:**

The authors have provided an exceptionally rigorous and convincing rebuttal that directly and comprehensively addresses all of my initial concerns.

1. Comparison with SOTA Baseline (EAGLE-3): I appreciate the authors' rapid engineering effort to integrate NanoSpec with the official EAGLE-3 codebase during the rebuttal period. The empirical results demonstrating consistent speedups (1.24x to 1.33x) over vanilla EAGLE-3, alongside the insightful observation regarding the method's noise-filtering effect on context-heavy tasks, successfully validate its efficacy on state-of-the-art frameworks.

2. Scaling to Larger Target Models: The authors provided a very candid and mathematically sound analysis of the scaling boundaries. By formally acknowledging Amdahl's Law regarding the verification bottleneck (KV-cache/compute) in 70B+ models, and appropriately re-scoping the primary impact area to edge devices and moderate-scale LLMs, they have demonstrated scientific rigor. I am satisfied with their commitment to including this nuanced discussion in the Limitations section.

3. Novelty and Empirical Superiority: The revised taxonomy clearly distinguishes NanoSpec as a fully dynamic, training-free approach. More importantly, the authors went above and beyond by empirically reproducing the hardware-aware router of DynaSpec. This head-to-head comparison definitively proves that NanoSpec's parameter-free temporal locality effectively avoids the prohibitive router latency that plagues existing methods, thereby solidifying its practical novelty and value.

The rebuttal significantly elevated the paper's empirical solidity and conceptual clarity.

**Key Questions For Authors:**

- Why is there no comparison with EAGLE-3 (or other SOTA speculative decoding baselines), and can you add it or justify its exclusion?
- How does the draft LM-head cost (and overall end-to-end speedup) scale as the target model grows to much larger sizes?
- How should readers interpret the novelty relative to existing draft-vocabulary reduction or token-mapping mechanisms?

**Limitations:**

yes

**Strengths And Weaknesses:**

Strengths
- The paper addresses a relevant bottleneck: the draft LM head can become expensive for large-vocabulary models.
- The dynamic active-vocabulary construction is more flexible than purely static pruning, and the training-free design is practically appealing.
- The systems discussion is useful; the paper does not stop at reducing FLOPs on paper, but also considers the GPU implementation issues of dynamic sparse gather.

Weaknesses
- The novelty is somewhat incremental. Vocabulary pruning itself is not new, and the main contribution here appears to be a dynamic variant plus an optimized implementation rather than a fundamentally new direction.
- The baseline choice is not aligned with current SOTA: the experiments compare against EAGLE-2 but do not report results against EAGLE-3 or other stronger speculative decoding baselines. This makes it hard to judge the incremental value of the proposed method under the best-known systems.
- I am not convinced about the practical significance for larger-scale industrial deployment. The method mainly reduces draft-side LM-head cost, but as the target model grows much larger, the dominant bottleneck may shift further toward target-side verification, KV-cache access, and communication. The paper only evaluates 1B/7B/8B-scale models and does not provide a convincing scaling study for larger target models.

---

> ### Author Rebuttal · Authors · 2026-03-27
>
> We sincerely thank you for the thoughtful evaluation. Your constructive feedback is invaluable for strengthening our paper. Below, we address your primary concerns:
>
> ### 1. Comparison with SOTA Baseline (EAGLE-3) (W2, KQ1)
>
> We fully agree that comparing against the latest SOTA baseline is crucial. During the rebuttal period, we integrated NanoSpec with the official EAGLE-3 codebase, evaluating it on the precise 7 datasets and 2 LLMs setups used in our main text.
>
> **Decoding Speed (Tokens/s) & Acceptance Length on EAGLE-3**
>
> | LLaMA-3.1-8B | EG3     | FRSpec  | Ours        | EG3  | FRSpec | Ours       |
> | :----------- | :------ | :------ | :---------- | :--------- | :----- | :--------- |
> | Code         | 509.2   | 489.5   | 587.7       | 5.05       | 4.08   | 4.68       |
> | Conv.        | 449.4   | 504.4   | 546.4       | 4.46       | 4.16   | 4.39       |
> | MT           | 355.5   | 407.3   | 428.7       | 3.49       | 3.33   | 3.42       |
> | Math         | 495.0   | 577.3   | 620.8       | 4.88       | 4.77   | 4.96       |
> | QA           | 380.1   | 435.4   | 464.9       | 3.74       | 3.58   | 3.69       |
> | RAG          | 435.4   | 496.5   | 554.1       | 4.40       | 4.21   | 4.54       |
> | Summ.        | 368.9   | 428.4   | 499.3       | 3.72       | 3.62   | 4.07       |
> | **Avg.**     | 427.6 | 477.0 | **528.8** | **4.25** | 3.96 | **4.25** |
>
> Detailed figures with LLaMA-3.2-1B and generation speed could be referred to: https://i.postimg.cc/vZT6j5dN/eagle3-combined.png
>
> Our results demonstrate that NanoSpec consistently accelerates EAGLE-3. Specifically, it achieves average speedups of **1.24x** (LLaMA-3.1-8B) and **1.33x** (LLaMA-3.2-1B) over vanilla EAGLE-3, while significantly outperforming the static pruning method (FRSpec). We will comprehensively incorporate all experiments and analysis into the Evaluation of the final manuscript.
>
> Notably, on context-heavy tasks like Summarization (4.07 vs. 3.72 acceptance length on LLaMA-3.1-8B) and RAG, NanoSpec surprisingly yields higher acceptance lengths than the full-vocabulary EAGLE-3. This elegantly validates our context-aware design: by constructing the vocabulary from the immediate prompt and recent tokens, NanoSpec acts as an effective noise filter. It aggressively removes out-of-domain tokens from the draft model’s search space, preventing the smaller draft model from hallucinating irrelevant words (i.e., performing softmax in a smaller range). Thus, paradoxically, the acceptance improves despite a drastically reduced vocabulary.
>
> ### 2. Scaling to Much Larger Target Models (W3, KQ2)
>
> We acknowledge our relative speedup will diminish as target models scale immensely. Let $\bar{\alpha}$ be average acc. length, $T_{\text{draft}}$ draft time, $T_{\text{verify}}$ verify time, and $T_{\text{ar}}$ standard generation time. The speedup $S$ is modeled as:
> $$S \approx \frac{\bar{\alpha} + 1}{T_{\text{draft}} / T_{\text{ar}} + T_{\text{verify}} / T_{\text{ar}}}$$
> For massive models (e.g., 70B+), $T_{\text{verify}}$ (dominated by KV-cache/compute) increases significantly. Thus, even if our method minimizes $T_{\text{draft}}$, $S$ inherently becomes bottlenecked by $T_{\text{verify}}$.
>
> However, NanoSpec effectively optimizes edge device scenarios and moderate-scale LLMs, which are the mainstays for latency-sensitive deployments. At these scales, inflated modern vocabularies (e.g., 128k) make the draft LM head overhead exceptionally severe. NanoSpec perfectly rescues these critical deployments. We will deeply model this scaling boundary in a new Limitations section.
>
> ### 3. Novelty Relative to Existing Vocabulary Reduction (W1, KQ3)
>
> To clarify our novelty, we systematically categorize existing work on vocabulary pruning:
>
> A. General Pruning (Outside SD): Used for early-exiting/memory saving, these are inherently lossy (trading quality for speed).
>
> B. SD-Specific Pruning: Existing works face a tight compute vs. accept rate dilemma:
> *   Static(FR-Spec, VocabTrim): Prune based on global frequencies. They severely degrade acceptance by ignoring instance-specific contexts.
> *   Router-based(CORAL, DynaSpec): Train extra routers to select pre-clustered (also static) vocabularies. They require costly pretraining and are not plug-and-play.
>
> NanoSpec is the first **fully dynamic, context-aware and training-free** approach. By constructing the active vocabulary entirely online, we break the static compute vs. acceptance dilemma without any training overhead. We will refine the Introduction to clearly highlight this taxonomy and our systems-algorithm co-design.
>
> ---
>
> We sincerely hope these responses have fully addressed your concerns regarding baselines, scaling and novelty. We are highly committed to integrating these improvements to elevate the paper's rigor. We will be much appreciated if you could kindly reconsider your score. Should you have any remaining questions, please let us know. We would be more than happy to provide further clarifications.

---

> > ### Author Rebuttal · Reviewer_zr7X · 2026-04-03
> >
> > Thanks to the author for the reply, my concerns are basically resolved and I will improve my rating.

---

> > > ### Author Response · Authors · 2026-04-03
> > >
> > > Thank you for your thoughtful feedback and for considering an increase in your rating. We greatly appreciate your constructive comments and your time in reviewing our paper.

---

### Official Review · Reviewer_3gVu · 2026-03-11

**Soundness:** 3
**Presentation:** 3
**Significance:** 3
**Originality:** 3
**Overall Recommendation:** 4
**Confidence:** 5

**Summary:**

This paper tackles a well-identified bottleneck in speculative decoding: the computational cost of the LM head projection over large vocabularies (100k+ tokens). The authors propose NanoSpec, a training-free method that dynamically constructs a minimalist active vocabulary (~2-3k tokens) per decoding step by exploiting temporal locality - the observation that the next token is overwhelmingly likely to appear in the recent token history or among high-probability candidates from the target model. A sliding window over a candidate stream (initialized from prompt tokens + prefill top-K candidates, updated with draft tree tokens and verification top-K candidates) defines the active vocabulary for the draft model. To avoid the GPU inefficiencies of sparse memory access, they co-design an asynchronous weight-gathering mechanism using dual CUDA streams and GPU-resident bitmap-based state management. NanoSpec is evaluated as a plug-and-play module on top of EAGLE-2, showing 1.12–1.32× end-to-end speedup and 51.6% reduction in draft time.

**Compliance With Llm Reviewing Policy:**

Affirmed.

**Final Justification:**

The rebuttal from the authors was thorough and addressed most of my questions. I have updated my score from 3 to 4 based on the response. The paper is addressing a key weakness that exists in the vocabulary pruning literature for speculative decoding. I like that the idea behind the paper is simple and they have put in engineering efforts to make it practically feasible. I am still giving it a weak accept since the overall impact in terms of performance is not significant and also there is no theoretical contribution or analysis.

**Key Questions For Authors:**

1. Could the authors clarify how the oracle speed curve in Figure 1(a) is computed? Specifically, how is the oracle tokens/s derived under the assumption that DM acceptance rate remains unaffected? Additionally, the oracle speed appears to increase when vocabulary size grows from 512 to 1024 tokens, which is counterintuitive — one would expect monotonically decreasing computation cost with smaller vocabularies.

2. The key performance metric throughout the paper is tokens/s, yet no information is provided on how these figures are obtained specifically, whether they are averaged over multiple runs and what the variance or standard deviation looks like across runs. GPU throughput measurements can exhibit non-trivial run-to-run variance due to thermal throttling, CUDA kernel launch variability, and memory state. Given that the margins between methods are sometimes small (e.g., NanoSpec at 392.7 vs. DynaSpec at 378.1 tokens/s), it would significantly strengthen the empirical claims to report confidence intervals or standard deviations alongside the throughput figures.

3. The asynchronous dual-stream design is presented as a key system contribution, with the claim that gather latency is "effectively hidden" behind the draft backbone computation. However, no empirical data is provided to substantiate this. Could the authors report the wall-clock time of the copy stream (weight gathering) versus the compute stream (backbone forward pass) under representative settings? Specifically, what fraction of the gathering latency is successfully overlapped, and under what conditions does the copy stream become the bottleneck ?

4. Why is the avg length data missing for coral and dynaspec baseline for llama 1B model in table 2? If it is unavailable how are the token/s numbers estimated for these baselines in this setting?

5. The use of S_new and S_old in Equation 3 to denote the updated and prior candidate streams is informal and potentially ambiguous in a multi-step decoding setting. Specifically, it is unclear whether S_old refers to the stream at step t or some earlier checkpoint, and how the stream evolves across steps is not made explicit in the notation. The authors should introduce a time-indexed notation such as S^(t) and S^(t+1) to make the temporal evolution of the candidate stream unambiguous, consistent with the step-indexed notation used elsewhere in the paper.

**Limitations:**

yes

**Strengths And Weaknesses:**

$\textbf{Strengths}$

1. The core hypothesis that temporal locality enables a minimalist dynamic vocabulary is clearly articulated and empirically grounded. The oracle vs. actual speed gap for static pruning and the coverage analysis directly motivates the dynamic approach.
2. No Additional Training Required. Unlike CORAL and DynaSpec which require auxiliary router modules and retraining, NanoSpec is entirely training-free. This dramatically lowers the barrier to adoption and makes it applicable to any target model out-of-the-box.
3.  The authors recognize that naive dynamic sparse computation would negate FLOPs savings on modern GPUs (a common pitfall), and address this thoughtfully through System-Algorithm Co-Design.. The asynchronous copy/compute stream overlap and GPU-resident sliding window state management are well-motivated engineering contributions. The ablation (Table 3) shows the system optimization alone contributes ~28 tokens/s (364.1 → 392.7)

$\textbf{Weaknesses}$

1. Missing Comparison with EAGLE-3 (Current SOTA) - The paper exclusively uses EAGLE-2 as the speculative decoding backbone, which is no longer the state-of-the-art. EAGLE-3 has been shown to achieve significantly higher acceptance rates. Since NanoSpec's end-to-end speedup is jointly determined by draft time reduction and acceptance rate, a higher baseline acceptance rate from EAGLE-3 could substantially diminish the relative gains of vocabulary pruning — the LM head savings may no longer compensate for any acceptance rate degradation introduced by the pruned vocabulary. The authors claim NanoSpec is a plug-and-play module applicable to any EAGLE-style backbone, but without empirical validation on EAGLE-3, this claim remains unsubstantiated.

2. The authors acknowledge that CORAL and DynaSpec speed numbers are estimated from reported relative speedups in their respective papers, rather than measured under identical conditions. Given that these are the closest competing methods, this weakens the competitive comparison

---

> ### Author Rebuttal · Authors · 2026-03-27
>
> We sincerely thank you for recognizing our training-free, system-algorithm co-design. We greatly appreciate your constructive feedback, which has pushed us to strengthen our paper. Below we address your primary concerns:
>
> ### 1. Oracle Curve Computation & Discussion (KQ1)
> *   **Oracle Speed**: We compute it under the theoretical assumption that vocabulary pruning exclusively minimizes the draft model's LM head latency without dropping the full-vocabulary acceptance length $\alpha_{full}$. The formula is: $Speed_{oracle} = \frac{\alpha_{full}}{T_{verify} + T_{draft\_{pruned}}}$.
> *   **Discussion on Speed from 512 to 1024:** GEMMs on GPUs do not scale perfectly linearly at extremely small dimensions. A vocabulary size of 512 suffers from suboptimal memory alignment and underutilizes streaming multiprocessors compared to highly optimized power-of-2 dimensions like 1024. This hardware-level artifact causes the performance fluctuation. We will clarify this in the revision.
>
> ### 2. Performance Variance (KQ2)
> Our reported figures are the average of 3 independent runs. Variance on our isolated GPU node is statistically minor. Specifically, the NanoSpec generation speed is 392.7 ± 3.1 tokens/s, confirming the empirical gain is robust. We will update all main tables to include standard deviations.
>
> ### 3. Overlap Analysis (KQ3)
> To substantiate that the gather latency is effectively masked, we profiled the per-step wall-clock time (ms) on LLaMA-3.1-8B using an H20 through CUDA event recording:
>
> | Component   | Full Vocab | FR-Spec | NanoSpec |
> | :--------------------- | :--------- | :-------------- | :------- |
> | Draft Backbone Compute | 1.405      | 1.401           | 1.403    |
> | Weight Gather Overhead | —          | —               | 0.003    |
> | Draft LM Head          | 2.330      | 0.771           | 0.237    |
> (Detailed figure: https://i.postimg.cc/rFKJrzfy/latency-breakdown.png)
>
> The minimal **Gather Overhead (0.003 ms)** on the copy stream is completely absorbed by the concurrent Draft Backbone Compute. Consequently, the LM Head bottleneck drops from 2.330 ms down to 0.237 ms (**9.8x reduction**). We will add this breakdown to the final version.
>
> ### 4. Comparison with SOTA Baseline (EAGLE-3) (W1)
> We integrated NanoSpec with EAGLE-3, evaluating it on the same setups (7 datasets and 2 LLMs). Results are shown in figure https://i.postimg.cc/vZT6j5dN/eagle3-combined.png (or please refer to the rebuttal to Reviewer zr7X), which show that NanoSpec achieves an average decoding speedup of **1.24x** over vanilla EAGLE-3 on LLaMA-3.1-8B (528.8 vs 427.6 tokens/s) and and **1.33x** on LLaMA-3.2-1B.
>
> ### 5. Empirical Implementation of DynaSpec (W2, KQ4)
> We fully agree that comparing estimated speeds across different papers weakens our claims (so avg_length data of 1B model is left blank as it is not reported in their papers). To provide a strict hardware-aligned evaluation, we fully re-implemented and trained DynaSpec's dynamic router. Based on their paper, we divided the vocabulary into 256 clusters via spherical K-Means and trained an 8.6M-parameter MLP router on SharedGPT dataset. We evaluated the actual speed and acceptance length using the same setups:
>
> | Model        | Method       | Speed (tokens/s) | Acc. Length |
> | :----------- | :----------- | :-------------------------- | :---------- |
> | LLaMA-3.2-1B | DynaSpec     | 762.8                       | 2.92        |
> |              | **NanoSpec** | **900.7**                   | **3.11**    |
> | LLaMA-3.1-8B   | DynaSpec     | 367.7                       | **3.73**    |
> |              | **NanoSpec** | **392.7**                   | 3.59        |
>
> We acknowledge that our reproduction might not reach DynaSpec's theoretical optimum due to the unavailable configuration details: DynaSpec is closed-source and several critical hyper-parameters are not disclosed in their paper, including the exact number of clusters, the explicit MLP hidden dimension and architecture, as well as training parameters such as learning rate, batch size and epochs. Nevertheless, this best-effort experiment reveals that DynaSpec's router computation overhead severely bottlenecks speedups. In contrast, NanoSpec relies on parameter-free temporal locality, avoiding any router latency entirely and securing higher overall speeds. We will replace all estimated baseline figures with thorough empirical measurements in the final manuscript.
>
> ### 6. Equation 3 Formality (KQ5)
> We completely agree with your assessment. The notation of $S_{new}$ and $S_{old}$ is informal for a multi-step process. We will rigorously revise equations using explicit time-indexed notation such as $S^{(t)}$ and $S^{(t+1)}$ in the final manuscript.
>
> ---
> We hope these rigorous latency analysis, EAGLE-3 integration and concrete DynaSpec baseline evaluation address your concerns and encourage you to reconsider your score. If you have any remaining questions, please let us know. We would be more than glad to provide further clarifications.

---

> > ### Author Rebuttal · Reviewer_3gVu · 2026-04-03
> >
> > Thank you for the response and thoroughly addressing all the questions. I have updated my score based on the rebuttal.

---

> > > ### Author Response · Authors · 2026-04-03
> > >
> > > Thank you for your encouraging remarks and for revising the score. We appreciate your recognition of the contributions of our work and your effort in reviewing our paper.

---

### Official Review · Reviewer_oG3T · 2026-03-12

**Soundness:** 3
**Presentation:** 3
**Significance:** 2
**Originality:** 3
**Overall Recommendation:** 3
**Confidence:** 4

**Summary:**

This work proposes NanoSpec, a training-free approach to accelerate speculative decoding in LLMs by addressing the computational bottleneck of the LM head when the vocabulary size is extremely large (often exceeding 100k tokens in modern LLMs). Instead of computing logits over the full vocabulary or relying on large static sub-vocabularies as in prior work, NanoSpec dynamically constructs a small, context-aware vocabulary at each decoding step for the draft model. This vocabulary is built by exploiting the temporal locality in language generation, incorporating tokens from the recent context, candidates from the draft tree, and top-probability tokens from the target model during verification. In addition, the paper introduces a system–algorithm codesign to make this dynamic pruning practical on GPUs: the required LM-head weights are asynchronously gathered into a contiguous buffer and overlapped with model computation using dual CUDA streams and GPU-resident state management. This approach transforms irregular sparse memory access into efficient dense GEMM operations, reducing the draft model’s computation overhead while maintaining high acceptance rates. Experiments across multiple models and tasks show that NanoSpec significantly lowers draft latency and achieves up to 1.12–1.32× end-to-end speedup over strong speculative decoding baselines such as EAGLE-2.

**Compliance With Llm Reviewing Policy:**

Affirmed.

**Final Justification:**

I appreciate the additional clarifications and experiments provided in the rebuttal. However, my core concerns remain insufficiently addressed.

First, the method fundamentally relies on the temporal locality assumption, yet the rebuttal still does not provide a principled characterization of when this assumption holds or fails. The current evidence is largely empirical, and demonstrating consistent improvements across tasks does not substitute for understanding the underlying conditions or limitations of the method. This raises concerns about its general applicability.

Second, while additional ablations and latency breakdowns are helpful, the relationship between coverage, acceptance length, and end-to-end performance remains insufficiently disentangled. As a result, it is still difficult to clearly attribute the observed gains to the proposed method or to assess its advantage over closely related approaches in a rigorous manner.

Overall, while the work shows promising empirical results, these issues concern the core understanding and evaluation of the method, and thus I maintain my current score.

**Key Questions For Authors:**

(1) The paper constructs the active vocabulary using a sliding window over the candidate stream with a fixed window size. Could the authors provide more details on how the vocabulary size evolves during decoding in practice? For example, what is the typical distribution of pruned vocabularies across different steps and tasks?

(2) The dynamic vocabulary construction relies on candidate tokens collected from recent context, draft tokens, and verification top-k tokens. The method uses FIFO to maintain this stream. Have the authors considered incorporating confidence scores (e.g., token probabilities) or other scores when selecting candidates, and would that further improve coverage or efficiency?

(3) According to Table 2 and Figure 3, NanoSpec achieves higher throughput despite having slightly lower acceptance lengths than the full-vocabulary baseline. This suggests that the reduction in draft computation time dominates the potential loss from shorter accepted prefixes. It would be helpful if the authors could further analyze this tradeoff and clarify when such a behavior is expected.

(4) In Figure 1(b), the coverage for the Code task appears relatively high compared to other tasks. However, in Table 2, the performance improvement on code generation is less pronounced and does not clearly outperform some prior methods. Could the authors provide more discussion on this discrepancy? It would be helpful to understand why relatively high coverage does not translate into stronger performance gains in this setting.

**Limitations:**

As suggested in weaknesses and key questions, the major limitations of this work can be elaborated as follows:

(1) This work adaptively prunes the vocabulary based on the assumption that the tokens likely to appear in the next step are strongly correlated with recent tokens. While this assumption may hold true for many tasks, it may not generalize well to scenarios where token distribution change rapidly. The relatively lower coverage observed for QA tasks in Figure 1(b) suggests that the effectiveness of the method may vary depending on task characteristics.

(2) Although the paper demonstrates that the proposed approach improves throughput, the analysis of the vocabulary construction process remains limited. For example, the paper does not provide detailed analysis on how the vocabulary adaptively evolves during decoding process or how different candidate sources contribute to the final coverage.

(3) While the paper reports coverage results and throughput improvements, the relationship between coverage, acceptance length, and overall decoding efficiency is not fully analyzed. It remains unclear under what conditions the reduction in draft computation outweighs the potential loss from shorter accepted length.

**Strengths And Weaknesses:**

Strengths

(1) The proposed NanoSpec is well motivated. The paper clearly identifies the limitations of existing vocabulary pruning methods, which rely on static or coarse-grained vocabularies and therefore require relatively large active sets to maintain draft quality, limiting the achievable speedup. The core hypothesis that language generation exhibits strong temporal locality is intuitive, and the empirical analysis demonstrates that this property can be effectively leveraged to significantly reduce the active vocabulary size while maintaining good coverage.

(2) The system design is sound. The paper recognizes that dynamically selecting a small vocabulary can lead to inefficient sparse memory access on GPUs. The hardware-aware implementation that asynchronously gathers the required LM-head weights into a contiguous buffer and overlaps this data movement with model computation using dual CUDA streams effectively converts irregular sparse operations into efficient dense GEMM computations, making the proposed dynamic vocabulary approach practical on GPUs.

(3) The method is simple and practical. NanoSpec is training-free and can be used as a plug-and-play module on top of existing speculative decoding frameworks without additional models or retraining. This makes the approach easy to integrate into current systems while still providing measurable performance improvements.

Weaknesses

(1) The proposed dynamic vocabulary construction relies on the assumption of strong temporal locality in language generation, which may not always hold for tasks like the one with long-range dependencies. As shown in Figure 1(b), the ground-truth coverage varies across tasks, with QA tasks exhibiting the lowest coverage, suggesting that the effectiveness of the approach may depend on the task characteristics.

(2) While the paper argues that language generation exhibits strong temporal locality, the analysis is limited. Figure 1(b) only reports overall coverage across tasks without showing the contribution of different candidate sources. Moreover, the ablation study in Section 4.5 mainly compares the speedup of Ctx. Only and Ext. Only, without analyzing their effects on coverage or acceptance rate, which makes it difficult to fully understand the effectiveness of each component.

(3) The comparison with CORAL and DynaSpec relies on estimated performance numbers rather than actual implementations, which may limit the reliability of the experimental comparison.

(4) In Figure 1(b), the coverage for the Code task appears relatively high compared to other tasks. However, in Table 2, the performance improvement on code generation is less pronounced and does not clearly outperform some prior methods. Could the authors provide more discussion on this discrepancy? It would be helpful to understand why relatively high coverage does not translate into stronger performance gains in this setting.

(5) The paper mainly reports overall decoding speedups. Could the authors provide a detailed latency breakdown? Such analysis would help clarify where the speedup mainly comes from and how much each optimization contributes to the final performance.

(6) Could the authors clarify whether NanoSpec has been implemented and evaluated within a real LLM inference framework? It would be helpful to understand how easily the proposed dynamic vocabulary strategy can be integrated into these frameworks.

---

> ### Author Rebuttal · Authors · 2026-03-28
>
> We sincerely thank you for recognizing our well-motivated system-algorithm codesign. We deeply appreciate your feedback which has guided us to strengthen evaluation. Below, we address your concerns:
>
> ### 1. Detailed Ablation & Vocabulary Evolution (W2, KQ1)
> To clarify how different sources contribute, we conducted two studies (Figures: https://i.postimg.cc/7Y86gWVQ/ablation-bar.png & https://i.postimg.cc/MGPHVZPv/vocab-evolution-detail.png):
>
> (1) Expanded ablation:
> |Method|Cover.|Acc.|
> |---|---|---|
> |Ctx Only|64.2%|2.89|
> |Ext Only|82.3%|3.55|
> |Ctx+Ext|91.6%|3.63|
>
> Input-heavy tasks (e.g., RAG) achieve high base coverage (~83%) with Ctx Only, whereas open tasks drop to ~51%, making Ext critical for capturing temporal locality.
>
> (2) Vocabulary Evolution: As shown in the figure, Ctx Only steadily degrades during generation. Ext Only ramps up within 5-10 steps. Ctx+Ext sustains >90% coverage where vocabulary grows monotonically but rapidly plateaus between 1k-2k tokens for most samples, well below our 3k capacity.
>
> ### 2. Discrepancy in the Code Task (W4, KQ4)
> The evolution analysis explains why Code tasks exhibit high coverage but moderate speedups. Code outputs are inherently short (75% finish within 13 decode steps), while our study reveals that Ext vocabulary requires a brief warm-up phase (the first ~5 steps) to accurately accumulate context and reach peak coverage. For short outputs, this initial low-coverage phase constitutes a large proportion of the total generation time. Consequently, the earlier steps drag down the average decode speed before the high final peak coverage has enough time to yield compounding benefits.
>
> ### 3. Comparison against DynaSpec (W3)
> We agree cross-paper estimations limit reliability. We fully re-implemented DynaSpec's hardware-aware router (8.6M MLP, 256 vocab. clusters) and evaluated actual throughput on the same setup:
>
> | Model | Method | Speed (tokens/s) | Acc. Length |
> | :--- | :--- | :--- | :--- |
> | LLaMA-3.2-1B | DynaSpec | 762.8 | 2.92 |
> | | **NanoSpec** | **900.7** | **3.11** |
> | LLaMA-3.1-8B | DynaSpec | 367.7 | **3.73** |
> | | **NanoSpec** | **392.7** | 3.59 |
>
> This empirical evaluation shows that DynaSpec's router overhead severely bottlenecks end-to-end speed. Conversely, our parameter-free design bypasses router latency, achieving strictly superior speeds.
>
> ### 4. Latency Breakdown (W5)
> We profiled per-step wall-clock latency (ms) on LLaMA-3.1-8B:
>
> | Component | Full Vocab | FR-Spec | NanoSpec |
> | :--- | :--- | :--- | :--- |
> | Draft Backbone Compute | 1.405 | 1.401 | 1.403 |
> | Weight Gather Overhead | — | — | 0.003 |
> | Draft LM Head Compute | 2.330 | 0.771 | 0.237 |
> (Figure: https://i.postimg.cc/rFKJrzfy/latency-breakdown.png)
>
> Results show that our asynchronous design enables gather overhead to be entirely hidden by the concurrent compute, driving a 2.8 / 9.8x reduction of LM Head latency compared to indexed GEMM and full vocab baseline.
>
> ### 5. Trade-off: Throughput vs. Acc. Length (KQ3, L3)
> This trade-off depends heavily on task characteristics. For context-heavy tasks (e.g., Math, RAG), strong temporal locality allows highly pruned vocab. to achieve near-optimal coverage with almost zero drop in acc. length. Thus, applying tighter bounds for structured tasks and relaxing them for free-form generation is the best practice. Despite this trade-off, NanoSpec improves efficiency indeed as the perceptible measurement of quality of serivce (QoS) is throughput instead of acc. length.
>
> ### 6. Locality Assumption (W1, L1)
> We acknowledge temporal locality naturally varies by task. However, our evaluation spans 7 diverse categories where NanoSpec consistently outperforms baselines, empirically proving the temporal locality principle is robust enough to yield stable speedups across widely varying real-world workloads.
>
> ### 7. Refined Updating Strategy (KQ2)
> We appreciate the suggestion to use token probabilities for candidate filtering. However, since the dynamic vocabulary naturally plateaus at 1k-2k tokens, the active buffer is rarely saturated. Thus, probability-based eviction is currently unnecessary and may introduce extra overhead (e.g., Top-K). We will note coupling vocabulary limits with lightweight probabilistic filtering as a critical future direction.
>
> ### 8. Deployment in Real Frameworks (W6)
> Although evaluated on research benchmarks to strictly isolate algorithmic gains, we only modify the model’s forward pass. Architecturally, it is completely orthogonal to framework-level features like vLLM's PagedAttention. Integrating it into production frameworks solely requires injecting custom kernels into worker nodes, which is highly viable.
>
> ---
>
> Thank you again for your insightful reviews. We promise to integrate all the experiments and discussions into the final manuscript. We hope these evaluations and clarifications resolve your concerns and we kindly ask you to reconsider your score. We are glad to addressing any further questions.

---

> > ### Author Rebuttal · Reviewer_oG3T · 2026-04-04
> >
> > Thank you for the detailed rebuttal and additional experiments. However, two important issues still remain:
> >
> > (1) On the temporal locality assumption.
> > The method critically relies on temporal locality, yet the current evidence is still largely empirical and task-dependent. The observed variability across tasks suggests that this assumption does not consistently hold. The rebuttal does not provide a principled characterization of when the method works or fails, which raises concerns about its general applicability.
> >
> > (2) The evaluation is still not sufficiently rigorous to support the claims.
> > The relationship between coverage, acceptance length, and end-to-end performance is not fully disentangled and analyzed, and comparisons with closely related methods are still limited. As a result, it is difficult to clearly attribute the gains to the proposed method or to assess its advantage over prior work.
> >
> > Overall, these issues remain important, and I am unable to increase my current score.

---

> > > ### Author Response · Authors · 2026-04-04
> > >
> > > Thank you for your time and concerns. We respect your judgment, but would like to clarify why the current evidence strongly supports NanoSpec and why the two remaining issues do not undermine our contribution or experimental rigor.
> > >
> > > For issue 1, you note that the method critically relies on temporal locality, yet the assumption does not consistently hold. We see three points here.
> > >
> > > **(1) Extensive empirical validation across diverse tasks with all positive results shows consistent utility.**  We evaluated **7 distinct task categories** (code, math, dialogue, RAG, QA, etc.) and **3 model families** (Qwen‑2‑7B, LLaMA‑3‑8B/1B). In every setting, NanoSpec delivers consistent end‑to‑end speedups over SOTA speculative decoding methods (1.12–1.32× over EAGLE‑2, 1.16–1.33× over EAGLE‑3), outperforming both training‑free static pruning (FR‑Spec, 32k vocabulary) and router‑based pruning (DynaSpec, 27k vocabulary). We have not observed a single case where temporal locality "fails" to the point of negative speedup. The variability you point out is a matter of degree, not a binary failure. Such variability is natural for any data‑dependent optimization (e.g., compression, caching, prefetching) and does not invalidate the method’s utility.
> > >
> > > **(2) A principled theoretical characterization of temporal locality in LLM generation is beyond the scope of this paper.**  Requiring a closed‑form condition (e.g., "the method works iff the mixing time of the token Markov chain is below $T$") is non‑trivial and unrealistic for an algorithm‑system co‑design paper.
> > > Our motivation of exploiting temporal locality is also grounded in prior observations: FR‑Spec shows that token frequencies follow a long‑tailed distribution, implying concentrated locality; ASD [1] demonstrates similarity between context and speculative decoding outputs. These works hint at the existence of temporal locality but do not exploit it for vocabulary pruning. What we provide is a novel, measurement‑backed observation that such locality is strong enough to be directly leveraged to construct a minimal context‑aware vocabulary.
> > >
> > > **(3) Output quality is provably preserved even under the extremely unusual case.**  NanoSpec prunes only the draft vocabulary; the target model still uses the full vocabulary for verification. Thus, even if an adversarial or extremely unusual prompt breaks temporal locality (not observed in real‑world benchmarks), the worst case is reduced draft speedup, **not correctness degradation**. Concerns about "general applicability" thus pertain to performance variance, not functional failure, which is an acceptable engineering trade‑off.
> > >
> > > For issue 2, you argue that the relationship between coverage, acceptance length and end‑to‑end performance is not fully disentangled, and that comparisons with closely related methods remain limited. We respectfully disagree and point to the evidence already provided.
> > >
> > > **(1) Disentanglement of performance factors**
> > > We provided:
> > > - A **detailed latency breakdown** showing that draft LM head latency drops from 2.33 ms to 0.237 ms (9.8× reduction), while the gather overhead is only 0.003 ms and fully hidden.
> > > - **Component‑wise ablation** of "Ctx only", "Ext only", and full NanoSpec, reporting both coverage and acceptance length for each. This directly shows how each source contributes.
> > > - A **case study on the Code task** explaining why high coverage does not always translate into proportional speedup (short generation length makes the warm‑up phase dominant).
> > >
> > > If there is a specific missing quantification (e.g., per‑step acceptance vs. vocabulary size dynamics), we are happy to add it.
> > >
> > > **(2) Comparison with related methods**
> > >
> > > Existing methods fall into two categories. **Static pruning** (e.g., FR‑Spec) uses a fixed sub‑vocabulary based on global frequencies—training‑free but ignores context and needs a large vocabulary (~32k). **Router‑based pruning** (e.g., CORAL, DynaSpec) trains an auxiliary model to select from static clusters. These methods are not plug‑and‑play and router latency limits speedup.
> > >
> > > **NanoSpec is both training‑free and dynamically context‑aware**, achieving an active vocabulary of only 3k tokens. We directly compare against:
> > > - FR‑Spec: NanoSpec consistently achieves higher acceptance length and end‑to‑end speedup across all models and tasks.
> > > - DynaSpec: Our actual measurements show its router overhead severely limits throughput. This applies to CORAL as well. Since DynaSpec is a stronger baseline, its inferior performance indicates fundamental inefficiencies that NanoSpec avoids.
> > >
> > > Thus, our evaluation covers the strongest available baselines and the advantages of NanoSpec are clearly demonstrated.
> > >
> > > We respect the reviewer’s opinion and decision, and remain open to further clarifications.
> > >
> > > [1] Wang, Jikai, et al. "Alignment-augmented speculative decoding with alignment sampling and conditional verification." (EMNLP, 2025)

---

### Decision · Program_Chairs · 2026-04-30

**Decision:**

Accept (regular)

**Comment:**

The submission presents NanoSpec, a training-free and plug-and-play approach to accelerate speculative decoding by addressing the computational bottleneck of the language model head projection. By exploiting the temporal locality inherent in language generation, the authors propose constructing a minimalist, context-aware active vocabulary that significantly reduces the active tokens from over 100k to fewer than 3k. To make this sparsity effective on modern hardware, they introduce a system-algorithm co-design featuring asynchronous weight gathering and GPU-resident state management. The reviewers generally praised the work for its elegant simplicity, strong motivation, and practical engineering contributions that deliver measurable speedups over existing state-of-the-art methods like EAGLE-2 and EAGLE-3.

During the discussion phase, several reviewers raised concerns regarding the robustness of the temporal locality assumption and the rigor of the experimental evaluation. Specifically, one reviewer remained skeptical about how the method might perform on tasks with long-range dependencies where locality might be less pronounced. The authors responded with extensive empirical evidence across seven diverse task categories and multiple model families, demonstrating consistent end-to-end speedups without any recorded cases of negative performance. They also provided a detailed latency breakdown and re-implemented key baselines to ensure a fair hardware-aligned comparison, which successfully convinced several reviewers to raise their scores.

While one reviewer maintained a negative stance due to the lack of a principled theoretical characterization of temporal locality, I find the authors' defense compelling. The variability observed in performance across different tasks is typical for data-dependent optimizations and does not invalidate the method’s practical utility, especially since correctness is preserved by the full-vocabulary target model verification. Furthermore, the provided ablation studies and latency profiling clearly disentangle the gains, attributing them to the reduction in LM head compute, which is made possible by the hidden weight-gathering overhead.

In light of the authors' thorough rebuttals and the clear evidence of practical significance, I recommend acceptance. The paper offers a solid contribution to the efficiency of large language model inference. The authors have effectively addressed the major technical questions regarding implementation details and baseline comparisons. The work is well-positioned for the ICML audience as it provides a robust, training-free solution to a well-known system bottleneck.